# Association analyses identify 31 new risk loci for colorectal cancer susceptibility

Philip J. Law, Maria Timofeeva, Ceres Fernandez-Rozadilla ⬤ et al.#

Colorectal cancer (CRC) is a leading cause of cancer-related death worldwide, and has a strong heritable basis. We report a genome-wide association analysis of 34,627 CRC cases and 71,379 controls of European ancestry that identifies SNPs at 31 new CRC risk loci. We also identify eight independent risk SNPs at the new and previously reported European CRC loci, and a further nine CRC SNPs at loci previously only identified in Asian populations. We use in situ promoter capture Hi-C (CHi-C), gene expression, and in silico annotation methods to identify likely target genes of CRC SNPs. Whilst these new SNP associations implicate target genes that are enriched for known CRC pathways such as Wnt and BMP, they also highlight novel pathways with no prior links to colorectal tumourigenesis. These findings provide further insight into CRC susceptibility and enhance the prospects of applying genetic risk scores to personalised screening and prevention.

---

Many colorectal cancers (CRC) develop in genetically susceptible individuals[1] and genome-wide association studies (GWAS) of CRC have thus far reported 43 SNPs mapping to 40 risk loci in European populations[2,3]. In Asians, 18 SNPs mapping to 16 risk loci have been identified[4,5], a number of which overlap with those reported in Europeans. Collectively across ethnicities GWAS has provided evidence for 53 unique CRC susceptibility loci. While much of the heritable risk of CRC remains unexplained, statistical modelling indicates that further common risk variants remain to be discovered[6].

To gain a more comprehensive insight into CRC aetiology, we conducted a GWAS meta-analysis that includes additional, unreported datasets. We examine the possible gene regulatory mechanisms underlying all GWAS risk loci by analysing in situ promoter Capture Hi-C (CHi-C) to characterise chromatin interactions between predisposition loci and target genes, examine gene expression data and integrate these data with chromatin immunoprecipitation-sequencing (ChIP-seq) data. Finally, we quantify the contribution of the loci identified in this study, together with previously identified loci to the heritable risk of CRC and estimate the sample sizes required to explain the remaining heritability.

## Results

**Association analysis**. Thus far, studies have identified 61 SNPs that are associated with CRC risk in European and Asian populations (Supplementary Data 1). To identify additional CRC risk loci, we conducted five new CRC GWAS, followed by a meta-analysis with 10 published GWAS totalling 34,627 cases and 71,379 controls of European ancestry under the auspices of the COGENT (COlorectal cancer GENeTics) consortium[7] (Fig. 1, Supplementary Data 2). Following established quality control measures for each dataset[8] (Supplementary Data 3), the genotypes of over 10 million SNPs in each study were imputed, primarily using 1000 Genomes and UK10K data as reference (see Methods). After filtering out SNPs with a minor allele frequency <0.5% and imputation quality score <0.8, we assessed associations between CRC status and SNP genotype in each study using logistic regression. Risk estimates were combined through an inverse-variance weighted fixed-effects meta-analysis. We found little evidence of genomic inflation in any of the GWAS datasets (individual $\lambda_{GC}$ values 1.01–1.11; meta-analysis $\lambda_{1000} = 1.01$, Supplementary Figure 1).

Excluding flanking regions of 500 kb around each previously identified CRC risk SNP, we identified 623 SNPs associated with CRC at genome-wide significance (logistic regression, $P < 5 \times 10^{-8}$). After implementing a stepwise model selection, these SNPs were resolved into 31 novel risk loci, with 27 exhibiting Bayesian False Discovery Probabilities (BFDPs)[9] <0.1 (Table 1, Fig. 2, Supplementary Figure 2). The association at 20q13.13 (rs6066825) had only been previously identified as significant in a multi-ethnic study[10]. Two new associations (rs3131043 and rs9271770) were identified within the 6p21.33 major histocompatibility (MHC) region, with rs3131043 located 470 kb 5′ of *HLA-C*, and rs9271770 located between *HLA-DRB1* and *HLA-DQA1*. Imputation of the MHC region using SNP2HLA[11] provided no evidence for additional MHC risk loci.

We confirmed 28 of the 40 risk loci for CRC published as genome-wide significant in Europeans (i.e. $P < 5 \times 10^{-8}$) (Supplementary Data 1). For four previously reported risk loci[2,12–14], we observed associations that were just below genome-wide significance (3q26.2, rs10936599, $P = 1.41 \times 10^{-7}$; 12p13.32, rs3217810, $P = 1.09 \times 10^{-6}$; 16q22.1, rs9929218, $P = 4.96 \times 10^{-7}$; 16q24.1, rs2696839, $1.28 \times 10^{-6}$). In contrast, there was limited support in our current study for eight of the associations previously reported

by others[2,10,15–17] (2q32.3, rs11903757, $P = 0.23$; 3p14.1, rs812481, $P = 0.44$; 4q22.2, rs1370821, $P = 3.41 \times 10^{-5}$; 4q26, rs3987, $P = 0.10$; 4q32.2, rs35509282, $P = 0.24$; 10q11.23, rs10994860, $P = 3.65 \times 10^{-4}$; 12q24.22, rs73208120, $P = 0.03$; 20q11.22, rs2295444, $P = 0.03$), all having a BFDP >0.99 (Supplementary Data 1). Of the 16 reported Asian-specific loci[4,5], nine harboured genome-wide significant signals in the current study (all BFDP <0.06), albeit sometimes at SNPs with low $r^2$ but high D′ with the original SNP in Europeans, consistent with differences in allele frequencies in the different populations (Supplementary Data 1). Conditioning on the reported Asian SNPs, five of the nine European risk SNPs were independent of the Asian SNP ($P_{conditional} < 5 \times 10^{-8}$, Supplementary Data 4). We found no evidence of association signals at the remaining previously reported Asian SNPs.

Next, we performed an analysis conditioned on the sentinel SNP ($r^2 < 0.1$ and $P_{conditional} < 5 \times 10^{-8}$; Table 2) to search for further independent signals at these new and previously reported risk loci. We confirmed the presence of previously reported dual signals at 14q22.2, 15q13.3 and 20p12.3[18]. For the new risk loci, an additional independent signal was identified at 5p15.3. In addition, a further seven signals were found at five previously reported risk loci: 11q13.4, 12p13.32, 15q13.3, 16q24.1, 20q13.13. Two of these signals were at the 15q13.3 locus, of which one was 5′ of *GREM1* and the other intronic to *FMN1*. A further two signals were proximal and distal of 20q13.13. At 12p13.32 and 16q24.1, genome-wide associations marked by rs12818766 and rs899244, respectively, were shown. These were independent of the previously reported associations[2,14] at rs3217810 and rs2696839 (pairwise $r^2 = 0.0$).

In total, we identified 39 new independent SNPs associated with CRC susceptibility at genome-wide significance in Europeans. Together with the nine associations previously identified in Asian populations, and the 31 previously identified SNPs that were confirmed here, this brought the number of identified CRC association signals in Europeans to 79. Several of these risk loci map to regions previously identified in other cancers. In particular, three regions harbour susceptibility loci for multiple cancers[19], specifically 5p15.33 (*TERT-CLPTM1L*), 9p21.3 (*CDKN2A*) and 20q13.33 (*RTEL1*) (Supplementary Data 5).

**Functional annotation and biological inference of risk loci**. To the extent that they have been deciphered, most GWAS risk loci map to non-coding regions of the genome influencing gene regulation[19]. Consistent with this, we found evidence that the CRC risk SNPs mapped to regions enriched for active enhancer marks (H3K4me1 and H3K27ac) in colonic crypts (permutation test, $P = 0.034$ and 0.033, respectively) and colorectal tumours ($P = 4.2 \times 10^{-3}$ and $4.0 \times 10^{-5}$) (Supplementary Figure 3). To determine whether the CRC SNPs overlapped with active regulatory regions in a cell-type specific manner[20], we analysed the H3K4me3, H3K27ac, H3K4me1, H3K27me3, H3K9ac, H3K9me3 and H3K36me3 chromatin marks across multiple cell types from the NIH Roadmap Epigenomics project[21]. Colonic and rectal mucosa cells showed the strongest enrichment of risk SNPs at active enhancer and promoter regions (H3K4me3, H3K4me1 and H3K27ac marks, $P < 5 \times 10^{-4}$) (Supplementary Figure 3).

Given our observation that the risk loci map to putative regulatory regions, we examined both histone modifications and transcription factor (TF) binding sites in LoVo and HT29 CRC cells across the risk SNPs. Using variant set enrichment[22], we identified regions of strong LD (defined as $r^2 > 0.8$ and D′ > 0.8) with each risk SNP and determined the overlap with ChIP-seq data from the Systems Biology of Colorectal cancer (SYSCOL) study and inhouse-generated histone data. We identified an over-representation of binding for MYC, ETS2, cohesin loading

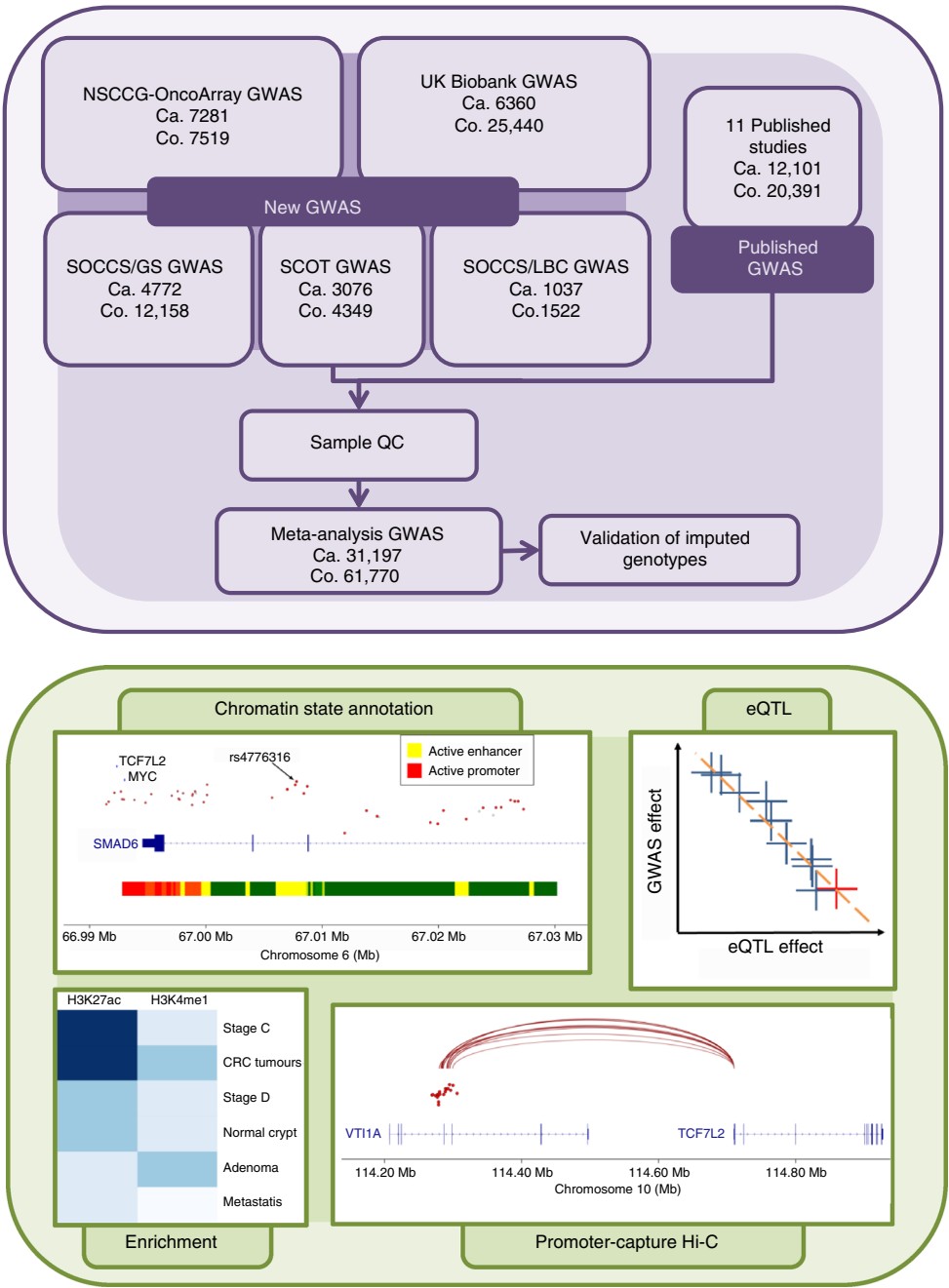

**Fig. 1** Study design

factor NIPBL and cohesin-related proteins RAD21, SMC1A and SMC3 (Supplementary Figure 4). About 87% (69/79) of the risk SNPs were predicted to disrupt binding motifs of specific TFs, notably CTCF, SOX and FOX, with 35% located within TF binding peaks from LoVo, HT29 or ENCODE ChIP-seq data (Supplementary Data 6).

The upstream mechanisms by which predisposition SNPs influence disease risk is often through effects on *cis*-regulatory transcriptional networks, specifically through chromatin-looping interactions that are fundamental for regulation of gene expression. Therefore, to link regulatory regions containing risk SNPs to promoters of candidate target genes, we applied in situ promoter capture Hi-C (CHi-C) data in LoVo and HT29 cells (Supplementary Data 9). About 38% of the risk SNPs mapped to regions that showed statistically significant chromatin-looping interactions with the promoters of respective target genes. Notably, as well

as confirming the interaction between rs6983267 and *MYC* at 8q24.21 (Supplementary Figure 2), the looping interaction from an active enhancer region at 10q25.2 implicates *TCF7L2* as the target gene of rs12255141 variation (Fig. 3). TCF7L2 (previously known as TCF4) is a key transcription factor in the Wnt pathway and plays an important role in the development and progression of CRC[23]. Intriguingly, TCF7L2 has been shown to bind to a *MYC* enhancer containing rs6983267[24] and to a *GREM1* enhancer near rs16969681[25]. Based on ChromHMM, this region is annotated as a promoter in HCT116 cells, but not in normal colonic and rectal mucosa. Additionally this locus has been implicated in lung cancer[26] and low-grade glioma[27]. Similarly, the 9p21.3 chromatin interaction provides evidence to support *CDKN2B* as the target gene for rs1412834 variation, a region of somatic loss.

We sought to gain further insight into the target genes at each locus, and hence the biological mechanisms for the associations,

**Table 1 Summary results for the new colorectal cancer risk loci in Europeans**

| SNP | Cytoband | Position (bp, GRCh37) | Risk/alt allele | RAF | OR | 95% CI | P-value | BFDP | $I^2$ (%) | $P_{het}$ | Average info score |
|---|---|---|---|---|---|---|---|---|---|---|---|
| rs61776719 | 1p34.3 | 38,461,319 | C/A | 0.45 | 1.07 | (1.05; 1.10) | $2.19 \times 10^{-10}$ | $1.98 \times 10^{-3}$ | 1 | 0.44 | 0.89 |
| rs12143541 | 1p32.3 | 55,247,852 | G/A | 0.15 | 1.10 | (1.06; 1.13) | $9.44 \times 10^{-10}$ | $7.44 \times 10^{-3}$ | 16 | 0.28 | 0.95 |
| rs11692435 | 2q11.2 | 98,275,354 | G/A | 0.90 | 1.12 | (1.07; 1.16) | $1.22 \times 10^{-8}$ | 0.079 | 29 | 0.14 | 0.97 |
| rs11893063 | 2q33.1 | 199,601,925 | A/G | 0.47 | 1.07 | (1.04; 1.09) | $9.34 \times 10^{-11}$ | 0.069 | 43 | 0.04 | 0.96 |
| rs7593422 | 2q33.1 | 200,131,695 | T/A | 0.55 | 1.07 | (1.05; 1.10) | $3.56 \times 10^{-11}$ | $3.50 \times 10^{-4}$ | 15 | 0.28 | 0.99 |
| rs9831861 | 3p21.1 | 53,088,285 | G/T | 0.59 | 1.07 | (1.05; 1.09) | $4.17 \times 10^{-10}$ | $3.72 \times 10^{-3}$ | 0 | 0.87 | 0.99 |
| rs12635946 | 3q13.2 | 112,916,918 | C/T | 0.62 | 1.08 | (1.06; 1.10) | $1.02 \times 10^{-11}$ | $1.03 \times 10^{-4}$ | 11 | 0.33 | 0.97 |
| rs17035294 | 4q24 | 106,048,291 | T/C | 0.83 | 1.10 | (1.07; 1.13) | $2.73 \times 10^{-10}$ | $2.30 \times 10^{-3}$ | 0 | 0.95 | 1.00 |
| rs75686861 | 4q31.21 | 145,621,328 | A/G | 0.10 | 1.12 | (1.08; 1.16) | $1.76 \times 10^{-9}$ | 0.014 | 0 | 0.49 | 0.92 |
| rs2070699 | 6p24.1 | 12,292,772 | T/G | 0.48 | 1.07 | (1.04; 1.09) | $3.88 \times 10^{-9}$ | 0.031 | 29 | 0.14 | 0.95 |
| rs3131043 | 6p21.33 | 30,758,466 | G/A | 0.43 | 1.07 | (1.05; 1.1) | $2.67 \times 10^{-8}$ | 0.159 | 60 | 0.01 | 0.91 |
| rs9271770 | 6p21.32 | 32,594,248 | A/G | 0.81 | 1.08 | (1.05; 1.11) | $3.60 \times 10^{-8}$ | 0.192 | 0 | 0.91 | 0.93 |
| rs6928864 | 6q21 | 105,966,894 | C/A | 0.91 | 1.13 | (1.09; 1.19) | $1.37 \times 10^{-8}$ | 0.094 | 0 | 0.73 | 0.98 |
| rs10951878 | 7p12.3 | 46,926,695 | C/T | 0.49 | 1.06 | (1.04; 1.09) | $1.10 \times 10^{-8}$ | 0.080 | 0 | 0.65 | 0.99 |
| rs3801081 | 7p12.3 | 47,511,161 | G/A | 0.68 | 1.08 | (1.06; 1.11) | $2.00 \times 10^{-11}$ | $1.96 \times 10^{-4}$ | 50 | 0.01 | 1.00 |
| rs1412834 | 9p21.3 | 22,110,131 | T/C | 0.50 | 1.08 | (1.06; 1.11) | $4.13 \times 10^{-14}$ | $5.05 \times 10^{-7}$ | 14 | 0.30 | 1.00 |
| rs4450168 | 11p15.4 | 10,286,755 | C/A | 0.17 | 1.10 | (1.06; 1.13) | $1.24 \times 10^{-8}$ | 0.079 | 0 | 0.81 | 0.86 |
| rs7398375 | 12q13.3 | 57,540,848 | C/G | 0.72 | 1.09 | (1.06; 1.13) | $3.91 \times 10^{-10}$ | $3.23 \times 10^{-3}$ | 0 | 0.93 | 0.83 |
| rs12427600 | 13q13.3 | 37,460,648 | C/T | 0.24 | 1.09 | (1.06; 1.11) | $5.43 \times 10^{-11}$ | $5.01 \times 10^{-4}$ | 0 | 0.81 | 0.99 |
| rs45597035 | 13q22.1 | 73,649,152 | A/G | 0.64 | 1.08 | (1.05; 1.10) | $2.16 \times 10^{-10}$ | $1.94 \times 10^{-3}$ | 0 | 0.53 | 0.96 |
| rs1330889 | 13q22.3 | 78,609,615 | C/T | 0.87 | 1.11 | (1.07; 1.14) | $6.50 \times 10^{-10}$ | $5.25 \times 10^{-3}$ | 0 | 0.59 | 0.97 |
| rs7993934 | 13q34 | 111,074,915 | T/C | 0.65 | 1.08 | (1.05; 1.10) | $3.03 \times 10^{-11}$ | $2.94 \times 10^{-4}$ | 0 | 0.55 | 0.99 |
| rs4776316 | 15q22.31 | 67,007,813 | A/G | 0.73 | 1.08 | (1.05; 1.10) | $1.11 \times 10^{-8}$ | 0.076 | 22 | 0.21 | 0.95 |
| rs10152518 | 15q23 | 68,177,162 | G/A | 0.19 | 1.08 | (1.05; 1.11) | $3.24 \times 10^{-8}$ | 0.180 | 0 | 0.84 | 0.97 |
| rs7495132 | 15q26.1 | 91,172,901 | T/C | 0.12 | 1.11 | (1.07; 1.14) | $7.92 \times 10^{-10}$ | $6.34 \times 10^{-3}$ | 29 | 0.14 | 0.99 |
| rs61336918 | 16q23.2 | 80,007,266 | A/T | 0.29 | 1.09 | (1.06; 1.12) | $2.04 \times 10^{-12}$ | $2.14 \times 10^{-5}$ | 0 | 0.90 | 0.99 |
| rs1078643 | 17p12 | 10,707,241 | A/G | 0.77 | 1.09 | (1.06; 1.12) | $4.14 \times 10^{-11}$ | $3.81 \times 10^{-4}$ | 0 | 0.56 | 0.92 |
| rs285245 | 19p13.11 | 16,420,817 | T/C | 0.11 | 1.11 | (1.07; 1.15) | $3.71 \times 10^{-8}$ | 0.195 | 2 | 0.42 | 0.91 |
| rs12979278 | 19q13.33 | 49,218,602 | T/C | 0.53 | 1.07 | (1.05; 1.09) | $6.11 \times 10^{-10}$ | $5.35 \times 10^{-3}$ | 15 | 0.28 | 0.96 |
| rs6066825 | 20q13.13 | 47,340,117 | A/G | 0.65 | 1.10 | (1.08; 1.13) | $3.82 \times 10^{-17}$ | $5.67 \times 10^{-10}$ | 0 | 0.49 | 0.99 |
| rs3787089 | 20q13.33 | 62,316,630 | C/T | 0.32 | 1.07 | (1.05; 1.10) | $5.80 \times 10^{-9}$ | 0.043 | 0 | 0.80 | 0.96 |
| **Associations previously only identified in Asian populations** | | | | | | | | | | | |
| rs639933 | 5q31.1 | 134,467,751 | C/A | 0.38 | 1.07 | (1.05; 1.10) | $1.14 \times 10^{-9}$ | $9.50 \times 10^{-3}$ | 0 | 0.73 | 0.98 |
| rs6933790 | 6p21.1 | 41,672,769 | T/C | 0.83 | 1.10 | (1.07; 1.14) | $3.65 \times 10^{-10}$ | $3.03 \times 10^{-3}$ | 21 | 0.23 | 0.91 |
| rs704017 | 10q22.3 | 80,819,132 | G/A | 0.60 | 1.10 | (1.08; 1.13) | $2.96 \times 10^{-16}$ | $4.15 \times 10^{-9}$ | 23 | 0.21 | 0.95 |
| rs12255141 | 10q25.2 | 114,294,892 | G/A | 0.10 | 1.11 | (1.07; 1.15) | $2.97 \times 10^{-9}$ | 0.022 | 0 | 0.81 | 0.96 |
| rs10849438 | 12p13.31 | 6,412,036 | G/T | 0.12 | 1.12 | (1.08; 1.16) | $1.04 \times 10^{-10}$ | $9.49 \times 10^{-4}$ | 21 | 0.23 | 0.95 |
| rs73975588 | 17p13.3 | 816,741 | A/C | 0.87 | 1.10 | (1.06; 1.13) | $8.71 \times 10^{-9}$ | 0.058 | 33 | 0.11 | 0.97 |
| rs9797885 | 19q13.2 | 41,873,001 | G/A | 0.71 | 1.08 | (1.05; 1.10) | $2.77 \times 10^{-10}$ | $2.43 \times 10^{-3}$ | 0 | 0.70 | 0.99 |
| rs6055286 | 20p12.3 | 7,718,045 | A/G | 0.15 | 1.11 | (1.07; 1.14) | $9.69 \times 10^{-11}$ | $8.61 \times 10^{-4}$ | 50 | 0.02 | 0.97 |
| rs2179593 | 20q13.12 | 42,660,286 | A/C | 0.72 | 1.07 | (1.05; 1.10) | $4.62 \times 10^{-9}$ | 0.035 | 0 | 0.67 | 0.97 |

BFDP calculated using prior = $10^{-5}$ and maximum relative risk = 1.2
*RAF* risk allele frequency in Europeans, *OR* odds ratio, *CI* confidence interval, *BFDP* Bayesian False Discovery Probability, $I^2$ proportion of the total variation due to heterogeneity, $P_{het}$ P-value for heterogeneity

by performing expression quantitative trait locus (eQTL) analysis in colorectal tissue. We analysed inhouse eQTL data generated from samples of normal colonic mucosa (INTERMPHEN study, $n = 131$ patients) and GTEx data from transverse colon ($n = 246$). For the previously identified risk loci, there were eQTLs for rs4546885 and *LAMC1* (1q25.3), rs13020391 and lnc-RNA *RP11–378A13.1* (2q35), and rs3087967 and *COLCA1, COLCA2* and *C11orf53* (11q23.1). Amongst the eQTL associations at the new risk loci, pre-eminent eQTLs were rs9831861 and *SFMBT1* (3p21.1), rs12427600 and *SMAD9* (13q13.3), and rs12979278 and *FUT2* and *MAMSTR* (19q13.33) (Supplementary Data 7). However, while multiple nominally significant *cis*-eQTLs were present, nearly half of all loci had no evidence of *cis*-eQTLs in the sample sets used.

In addition to eQTL analysis, we performed Summary-data-based Mendelian Randomization (SMR) analysis[28] as a more stringent test for causal differences in gene transcription (Supplementary Data 8). There was support for the 11q23.1 locus SNP influencing CRC risk through differential expression of one or more of *COLCA1, COLCA2* and *C11orf53* transcripts ($P_{SMR} < 10^{-10}$). There was also evidence that the 3p21.1 and 19q13.33 SNPs acted through *SFMBT1* and *FUT2*, respectively, ($P_{SMR} < 10^{-5}$), as well as the 6p21.31 SNP acted through class II *HLA* expression ($P_{SMR} < 5 \times 10^{-4}$).

Based on genetic fine-mapping and functional annotation, our data indicated several candidate target genes with functions previously unconnected to colorectal tumourigenesis (Supplementary Data 9). The SFMBT1 protein (3p21.1) acts as a histone reader and a component of a transcriptional repressor complex[29]. *TNS3* at 7p12.3 encodes the focal adhesion protein TENSIN3, to which the intestinal stem cell marker protein Musashi1 has been reported to bind. Tns3-null mice exhibit impaired intestinal epithelial development, probably because of defects in Rho GTPase signalling and cell adhesion[30]. LRP1 (12q13.3, LDL receptor-related protein 1) (Fig. 3) may be involved in Wnt-signalling[31], although its role in the intestines has not previously been conclusively demonstrated. *FUT2* at 19q13.33 encodes fucosyltransferase II. Variation at this locus is associated with differential interactions with intestinal bacteria and viruses. Our data thus provide evidence for a role of the microbiome in CRC risk[32]. *PTPN1* (20q13.13), also known as *PTP1B*, encodes a non-receptor tyrosine phosphatase involved in regulating JAK-signalling, IR, c-Src, CTNNB1, and EGFR.

We annotated all risk loci with five types of functional data: (i) presence of a CHi-C contact linking to a gene promoter, (ii) presence of an association from eQTL, (iii) presence of a regulatory state, (iv) evidence of TF binding, and (v) presence of a nonsynonymous coding change (Supplementary Data 9). Collectively this analysis suggested three primary candidate disease mechanisms across a number of risk loci: firstly, genes linked to BMP/TGF-β signalling (e.g. *GREM1, BMP2, BMP4, SMAD7, SMAD9*); secondly, genes with roles either directly or indirectly linked to MYC (e.g. *MYC, TCF7L2*); and thirdly, genes with roles in maintenance of chromosome integrity (e.g.

*TERT*, *RTEL1*) and DNA repair (e.g. *POLD3*) (Supplementary Figure 5).

Pathway gene set enrichment analyses[33] revealed several growth or development related pathways were enriched, notably TGF-β signalling and immune response pathways (Supplementary Figure 6, Supplementary Data 10). Other cancer-related themes included apoptosis and leukocyte differentiation pathways. We used Data-driven Expression-Prioritized Integration for Complex Traits (DEPICT)[34] to predict gene targets based on gene functions that are shared across genome-wide significant risk loci, as well as those associated at $P < 10^{-5}$ as advocated to mitigate against type II error. Tissue-specificity with respect to colonic tissue was evident (permutation test, $P < 5 \times 10^{-3}$) and among the protein-coding genes predicted, there was enrichment for TGF-β and PI3K-signalling pathways, and abnormal intestinal crypt gene sub-networks ($P < 10^{-5}$; Supplementary Data 11).

**Contribution of risk SNPs to heritability**. Using Linkage Disequilibrium Adjusted Kinships (LDAK)[35] in conjunction with the GWAS data generated on unselected CRC cases (i.e. COIN, CORSA, Croatia, DACHS, FIN, SCOT, Scotland1, SOCCS/LBC, SOCCS/GS, UKBB, VQ58 studies) we estimated that the heritability of CRC attributable to all common variation is 0.29 (95% confidence interval: 0.24–0.35). To estimate the sample size required to explain a greater proportion of the GWAS heritability, we implemented a likelihood-based approach using association statistics in combination with LD information to model the effect-size distribution[36], which was best represented by a three-component model (mixture of two normal distributions). Under this model, to identify SNPs explaining 80% of the GWAS heritability, it is likely to require effective sample sizes in excess of 300,000 if solely based on GWAS associations (Supplementary Figure 7).

After adjusting for winner's curse[37], the 79 SNPs thus far shown to be associated with CRC susceptibility in Europeans explain 11% of the 2.2-fold familial relative risk (FRR)[38], whilst all common genetic variants identifiable through GWAS could explain 73% of the FRR. Thus, the identified susceptibility SNPs collectively account for approximately 15% of the FRR of CRC that can be explained by common genetic variation. We incorporated the newly identified SNPs into risk prediction models for CRC and derived a polygenic risk score (PRS) based on a total of 79 GWAS significant risk variants. Individuals in the top 1% have a 2.6-fold increased risk of CRC compared with the population average (Supplementary Figure 8). Risk reclassification using this PRS offers the prospect of optimising prevention programmes for CRC in the population, for example through targeting screening[6], and also preventative interventions. The identification of further risk loci through the analysis of even larger GWAS is likely to improve the performance of any PRS model.

**Co-heritability with non-cancer traits**. We implemented cross-trait LD score regression[39] to investigate co-heritability globally between CRC and 41 traits with publicly available GWAS summary statistics data. None of the genetic correlations remained significant after Bonferroni correction (two-sided Z-test, P-threshold: $0.05/41 = 1.2 \times 10^{-3}$). However, nominally significant positive associations with CRC risk (Supplementary Data 12) included insulin resistance, comprising raised fasting insulin, glucose and HbA1c (positive), hyperlipidaemia, comprising raised total cholesterol and low-density lipoprotein cholesterol, and ulcerative colitis, all of which are traits or diseases previously reported in observational epidemiological studies to be associated with CRC risk[40,41].

## Discussion

Here we report a comprehensive analysis that sheds new light on the molecular basis of genetic risk for a common cancer, and greatly increases the number of known CRC risk SNPs. To identify the most credible target genes at each site, we have performed detailed annotation using public databases, and have also acquired our own disease-specific data from ChIP-seq, promoter capture Hi-C and gene expression analyses.

Given that there remains significant missing common heritability for CRC, additional GWAS meta-analyses are likely to lead to discovery of more risk loci. Such an assertion is directly supported a contemporaneous study[42], which has reported the identification of 40 independent signals; 30 novel loci and 10 conditionally independent association signals at previously and newly identified CRC risk loci. Of these, 18 were replicated in our analysis, with an additional five exhibiting an independent signal present at the same locus (Supplementary Data 13).

Overall, our findings provide new insights into the biological basis of CRC, not only confirming the importance of established gene networks, but also providing evidence that point to a role for the gut microbiome in CRC causation, and identifying several functional mechanisms previously unsuspected of any involvement in colorectal tumourigenesis. Several of the gene pathways identified through GWAS may provide potential novel targets for chemoprevention and chemotherapeutic intervention.

## Methods

**Ethics**. Collection of patient samples and associated clinico-pathological information was undertaken with written informed consent and relevant ethical review board approval at respective study centres in accordance with the tenets of the Declaration of Helsinki. Specifically: (i) UK National Cancer Research Network Multi-Research Ethics Committee (02/0/097 [NSCCG], 01/0/5) [SOCCS], 05/S1401/89 [GS:SFHS], LREC/1998/4/183 [LBC1921], 2003/2/29 [LBC1936], 17/SC/0079 [CORGI] and 07/S0703/136 [SCOT]); (ii) The research activities of UK Biobank were approved by the North West Multi-centre Research Ethics Committee (11/NW/0382) in relation to the process of participant invitation, assessment and follow-up procedures. Additionally, ethics approvals from the National Information Governance Board for Health & Social Care in England and Wales and approval from the Community Health Index Advisory Group in Scotland were also obtained to gain access to the information that would allow the invitation of participants. This study did not need to re-contact the participants, and no separate ethics approval was required according to the Ethics and Governance Framework (EGF) of UK Biobank; (iii) South East Ethics Committee MREC (03/1/014); (iv) Written informed consent was obtained from all participants of CORSA. The study was approved by the ethical review committee of the Medical University of Vienna (MUW, EK Nr. 703/2010) and the "Ethikkommission Burgenland" (KRAGES, 33/2010) and (v) Finnish National Supervisory Authority for Welfare and Health, National Institute for Health and Welfare (THL/151/5.05.00/2017), the Ethics Committee of the Hospital District of Helsinki and Uusimaa (HUS/408/13/03/03/09).

The diagnosis of colorectal cancer (ICD-9 153, 154; ICD-10 C18.9, C19, C20) was established in all cases in accordance with World Health Organization guidelines.

**Primary GWAS**. We analysed data from five primary GWAS (Supplementary Data 2 and Supplementary Data 3):

(1) The NSCCG-OncoArray GWAS comprised 6240 cases ascertained through the National Study of Colorectal Cancer Genetics (NSCCG)[43] and 1041 cases collected through the CORGI consortium, genotyped using the Illumina OncoArray. Patients were selected for having a family history of CRC (at least one first-degree relative) or age of diagnosis below 58. Controls were also genotyped using the OncoArray and comprised (i) 3031 cancer-free men recruited by the PRACTICAL Consortium—the UK Genetic Prostate Cancer Study (UKGPCS) (age <65 years), a study conducted through the Royal Marsden NHS Foundation Trust and SEARCH (Study of Epidemiology & Risk Factors in Cancer), recruited via GP practices in East Anglia (2003–2009) and (ii) 4,488 cancer-free women across the UK, recruited via the Breast Cancer Association Consortium (BCAC).

(2) The SCOT GWAS comprised 3076 cases from the Short Course Oncology Treatment (SCOT) trial—a study of adjuvant chemotherapy in colorectal cancer by the CACTUS and OCTO groups[44]. Controls comprised 4349 cancer-free individuals from The Heinz Nixdorf Recall study[45]. Both cases and controls were genotyped using the Illumina Global Screening Array.

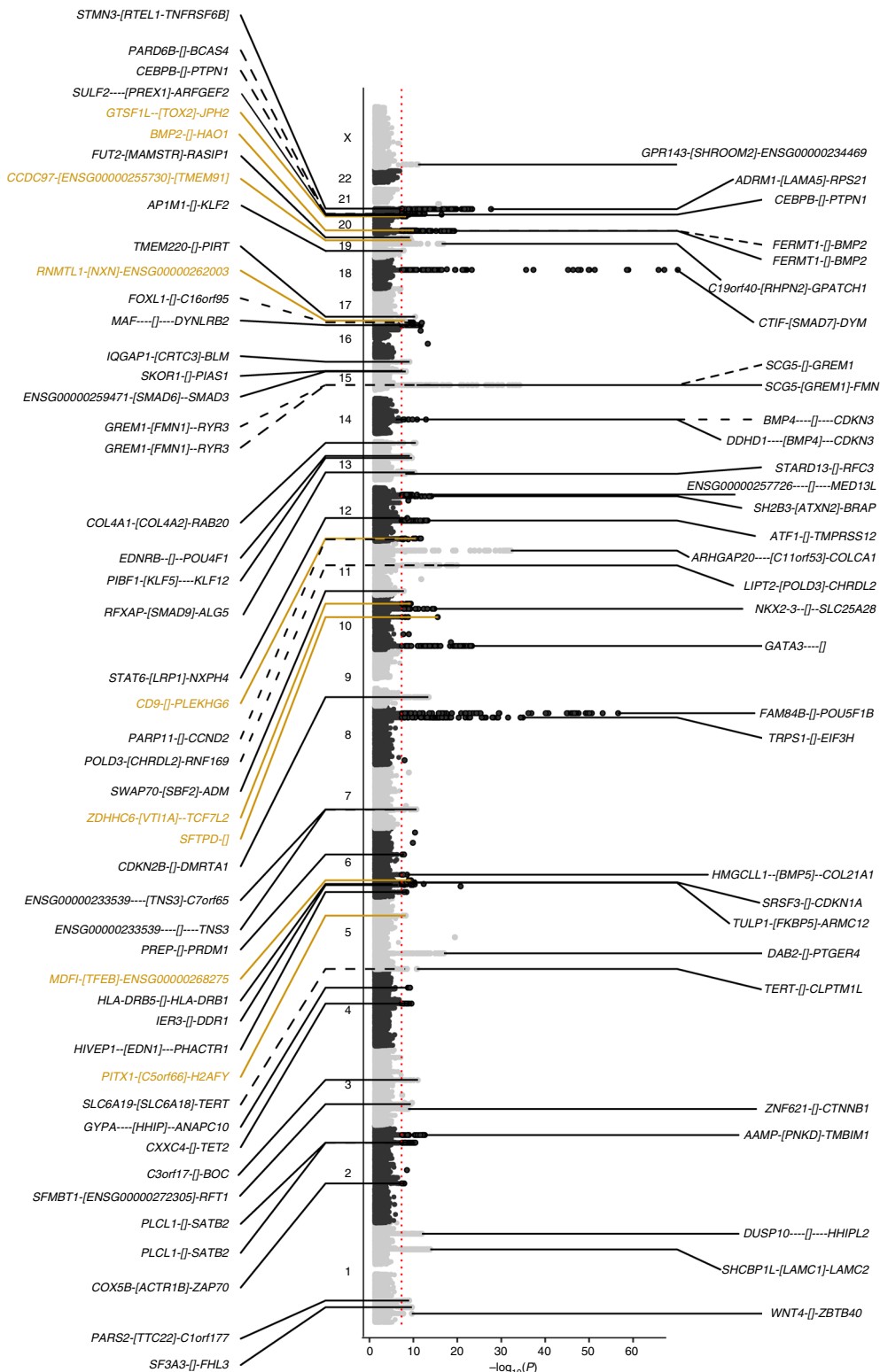

**Fig. 2** Manhattan plot showing all loci containing genetic risk variants independently associated with colorectal cancer risk at $P < 5 \times 10^{-8}$ in European populations. SNPs on the left of the plot are new SNPs identified in this study, and SNPs on the right were identified in previous studies and replicated at genome-wide significance in this study. The 79 risk SNPs consisted of 31 previously reported SNPs, 39 new risk SNPs, and nine SNPs previously identified in Asian but not in European populations (denoted in dark gold). Dotted lines indicate SNPs that were identified as independent through conditional analysis. Square brackets indicate the position of the sentinel SNP relative to nearest genes ("gene1-[]-gene2" for intergenic SNPs, "[gene]" for intragenic SNPs). The distance from the sentinel SNP to each gene is proportional to the number of dashes. The red line indicates the genome-wide significance threshold. The x-axis represents the $-\log_{10}P$-values of the SNPs, and the y-axis represents the chromosomal positions

**Table 2 Colorectal cancer variants identified in analysis conditioning on the sentinel SNP at each risk locus**

| Conditional (Sentinel) SNPs | Cytoband (position (bp, GRCh37)) | Risk/ Alt Allele | RAF | OR (95% CI) | P-value | Conditional OR (95% CI) | Conditional P-value | BFDP | LD with sentinel SNP ($r^2$; D') | $I^2$ (%) | $P_{het}$ | Average info score |
|---|---|---|---|---|---|---|---|---|---|---|---|---|
| rs77776598 (rs2735940) | 5p15.33 (1,240,998) | C/T | 0.06 | 1.14 (1.09;1.20) | $7.90 \times 10^{-9}$ | 1.16 (1.11;1.21) | $2.84 \times 10^{-10}$ | 0.003 | 0.00; 0.33 | 0 | 0.93 | 0.99 |
| rs4944940 (rs3824999) | 11q13.4 (74,415,252) | G/A | 0.96 | 1.31 (1.24;1.39) | $1.05 \times 10^{-20}$ | 1.28 (1.21;1.35) | $3.21 \times 10^{-17}$ | $2.73 \times 10^{-9}$ | 0.00; 0.19 | 6 | 0.38 | 0.95 |
| rs12818766 (rs3217810) | 12p13.32 (4,376,091) | A/G | 0.18 | 1.10 (1.07;1.13) | $2.15 \times 10^{-9}$ | 1.10 (1.07;1.13) | $5.29 \times 10^{-9}$ | 0.037 | 0.00; 0.06 | 30 | 0.16 | 0.89 |
| rs1570405[a] (rs4444235) | 14q22.2 (54,554,234) | G/A | 0.31 | 1.06 (1.03;1.08) | $9.81 \times 10^{-7}$ | 1.07 (1.04;1.09) | $1.91 \times 10^{-8}$ | 0.125 | 0.02; 0.19 | 0 | 0.46 | 1.00 |
| rs16969681[b] (rs73376930) | 15q13.3 (32,993,111) | T/C | 0.09 | 1.22 (1.18;1.27) | $2.97 \times 10^{-27}$ | 1.21 (1.16;1.25) | $2.85 \times 10^{-24}$ | $1.33 \times 10^{-16}$ | 0.01; 0.32 | 42 | 0.04 | 0.99 |
| rs16959063 (rs73376930) | 15q13.3 (33,105,730) | A/G | 0.01 | 1.30 (1.18;1.42) | $3.72 \times 10^{-8}$ | 1.33 (1.21;1.45) | $5.40 \times 10^{-9}$ | 0.23 | 0.00; 0.40 | 30 | 0.13 | 0.96 |
| rs17816465 (rs73376930) | 15q13.3 (33,156,386) | A/G | 0.20 | 1.11 (1.08;1.14) | $1.12 \times 10^{-14}$ | 1.12 (1.09;1.15) | $8.36 \times 10^{-15}$ | $1.07 \times 10^{-7}$ | 0.00; 0.11 | 44 | 0.04 | 0.97 |
| rs899244 (rs2696839) | 16q24.1 (86,700,030) | T/C | 0.21 | 1.09 (1.06;1.12) | $1.11 \times 10^{-10}$ | 1.09 (1.06;1.12) | $1.13 \times 10^{-10}$ | $4.06 \times 10^{-3}$ | 0.00; 0.04 | 14 | 0.29 | 0.99 |
| rs6085661[c] (rs961253) | 20p12.3 (6,693,128) | T/C | 0.39 | 1.09 (1.06;1.11) | $1.63 \times 10^{-14}$ | 1.09 (1.07;1.11) | $2.95 \times 10^{-15}$ | $3.88 \times 10^{-8}$ | 0.00; 0.08 | 0 | 0.96 | 1.00 |
| rs4811050 (rs1810502) | 20q13.13 (48,980,670) | A/G | 0.18 | 1.10 (1.07;1.13) | $2.43 \times 10^{-11}$ | 1.09 (1.06;1.12) | $4.07 \times 10^{-9}$ | $4.06 \times 10^{-3}$ | 0.04; 0.45 | 20 | 0.23 | 0.99 |
| rs6091213 (rs1810502) | 20q13.13 (49,384,745) | C/T | 0.26 | 1.08 (1.05;1.11) | $4.35 \times 10^{-10}$ | 1.08 (1.05;1.11) | $5.68 \times 10^{-10}$ | $4.76 \times 10^{-3}$ | 0.00; 0.05 | 6 | 0.39 | 0.94 |

BFDP calculated using prior = $10^{-5}$ and maximum relative risk = 1.2. LD calculated based on European populations in the 1000 Genomes Project data. BFDP calculated using conditional analysis results, with prior = $10^{-5}$ and maximum relative risk = 1.2

RAF risk allele frequency, OR odds ratio, CI confidence interval, BFDP Bayesian False Discovery Probability, $I^2$ proportion of the total variation due to heterogeneity, $P_{het}$ P-value for heterogeneity
[a]Tags to rs1957636 ($r^2 = 0.67$, D' = 1). Previously identified in Tomlinson IP, Nat Genet, 2008 (PMID:18372905).
[b]Previously identified in Tomlinson IP, Nat Genet, 2008 (PMID:18372905)
[c]Tags to rs4813802 ($r^2 = 0.75$, D' = 0.93). Previously identified in Tomlinson IP, Nat Genet, 2008 (PMID:18372905)

(3) SOCCS/Generation Scotland (SOCCS/GS) comprised 4772 cases from the Study of Colorectal Cancer in Scotland (SOCCS)[12,13] and 12,158 controls including 2221 population-based controls from SOCCS and additional 9937 population controls without prior history of colorectal cancer from Generation Scotland-Scottish Family Health Study (GS:SFHS)[46].

(4) SOCCS/Lothian Birth Cohort (SOCCS/LBC) GWAS comprised 1037 cases from the Study of Colorectal Cancer in Scotland (SOCCS)[47] and 1522 population-based controls without prior history of malignant tumours from the Lothian Birth Cohorts (LBC) of 1921 and 1936[48].

(5) UK Biobank (UKBB) GWAS comprised 6360 cases and 25,440 population-based control individuals. UK Biobank is a large cohort study with more than 500,000 individuals recruited. Biological samples of these participants were genotyped using the custom-designed Affymetrix UK BiLEVE Axiom array on an initial 50,000 participants and Affymetrix UK Biobank Axiom array on the remaining 450,000 participants. The two arrays had over 95% common content. Genotyping was done at the Affymetrix Research Services Laboratory in Santa Clara, California, USA. Details on genotyping and quality control were previously reported[49]. Self-reported cases of cancers of bowel, colon or rectum, if not confirmed by the ICD9 or ICD10 codes were excluded from the analysis. Healthy control individuals without history of cancer and/or colorectal adenoma were included in the analysis after matching one case to four controls by age, gender, date of blood draw, ethnicity and region of residence (two first letters of postal code).

**Published GWAS**. We made use of 10 previously published GWAS (Supplementary Data 2): (1) UK1 (CORGI study) comprised 940 cases with colorectal neoplasia and 965 controls[12]; (2) Scotland1 (COGS study) included 1012 CRC cases and 1012 controls[12]; (3) VQ58 comprised 1800 cases from the UK-based VICTOR and QUASAR2 adjuvant chemotherapy clinical trials and 2690 population control genotypes from the Wellcome Trust Case Control Consortium 2 (WTCCC2) 1958 birth cohort[50]; (4) CCFR1 comprised 1290 familial CRC cases and 1055 controls from the Colon Cancer Family Registry (CCFR)[15]; (5) CCFR2 included a further 796 cases from the CCFR and 2236 controls from the Cancer Genetic Markers of Susceptibility (CGEMS) studies of breast and prostate cancer[51,52]; (6) COIN was based on 2244 CRC cases ascertained through two independent Medical Research Council clinical trials of advanced/metastatic CRC (COIN and COIN-B)[53] and controls comprised 2162 individuals from the UK Blood Service Control Group genotyped as part of the WTCCC2; (7) Finnish GWAS (FIN)[3] was based on 1172 CRC cases and 8266 cancer-free controls

ascertained through FINRISK, Health 2000, Finnish Twin Cohort and Helsinki Birth Cohort Studies; (8) CORSA (COloRectal cancer Study of Austria) a molecular epidemiological study of 978 cases and 855 colonoscopy-negative controls[54]; (9) DACHS (Darmkrebs: Chancen der Verhütung durch Screening)[55] based on 1105 cases and 700 controls and (10) Croatia consisted of 764 cases and 460 population-based controls[56].

The VQ58, UK1 and Scotland1 GWAS were genotyped using Illumina Hap300, Hap240S, Hap370, Hap550 or Omni2.5 M arrays. 1958BC genotyping was performed as part of the WTCCC2 study on Hap1.2M-Duo Custom arrays. The CCFR samples were genotyped using Illumina Hap1M, Hap1M-Duo or Omni-express arrays. CGEMS samples were genotyped using Illumina Hap300 and Hap240 or Hap550 arrays. The COIN cases were genotyped using Affymetrix Axiom Arrays and the Blood Service controls were genotyped using Affymetrix 6.0 arrays. FIN cases were genotyped using Illumina HumanOmni 2.5M8v1 and controls using Illumina HumanHap 670k and 610k arrays. DACHS study samples were genotyped using the Illumina OncoArray, CORSA study sampels were genotyped on the Affymetrix Axiom Genome-Wide CEU 1 Array, and Croatia study samples were genotyped on Illumina OmniExpressExome BeadChip 8v1.1 or 8v1.3.

**Quality control**. Standard quality control (QC) measures were applied to each GWAS[8]. Specifically, individuals with low SNP call rate (<95%) as well as individuals evaluated to be of non-European ancestry (using the HapMap version 2 CEU, JPT/CHB and YRI populations as a reference) were excluded (Supplementary Figure 9). For apparent first-degree relative pairs, we excluded the control from a case-control pair; otherwise, we excluded the individual with the lower call rate. SNPs with a call rate <95% were excluded as were those with a MAF <0.5% or displaying significant deviation from Hardy–Weinberg equilibrium ($P < 10^{-5}$). QC details are provided in Supplementary Data 3. All genotype analyses were performed using PLINK v1.9[57].

**Imputation and statistical analysis**. Prediction of the untyped SNPs was carried out using SHAPEIT v2.837[58] and IMPUTE v2.3.2[59]. The CCFR1, CCFR2, COIN, CORSA, Croatia, NSCCG-OncoArray, SCOT, Scotland1, SOCCS/GS, SOCCS/LBC, UK1 and VQ58 samples used a merged reference panel using data from 1000 Genomes Project (phase 1, December 2013 release) and UK10K (April 2014 release). Imputation of UKBB was based on data from 1000 Genomes Project (phase 3), UK10K and Haplotype Reference Consortium. The FIN and DACHS GWAS were imputed using a reference panel comprised of 1000 Genomes Projects

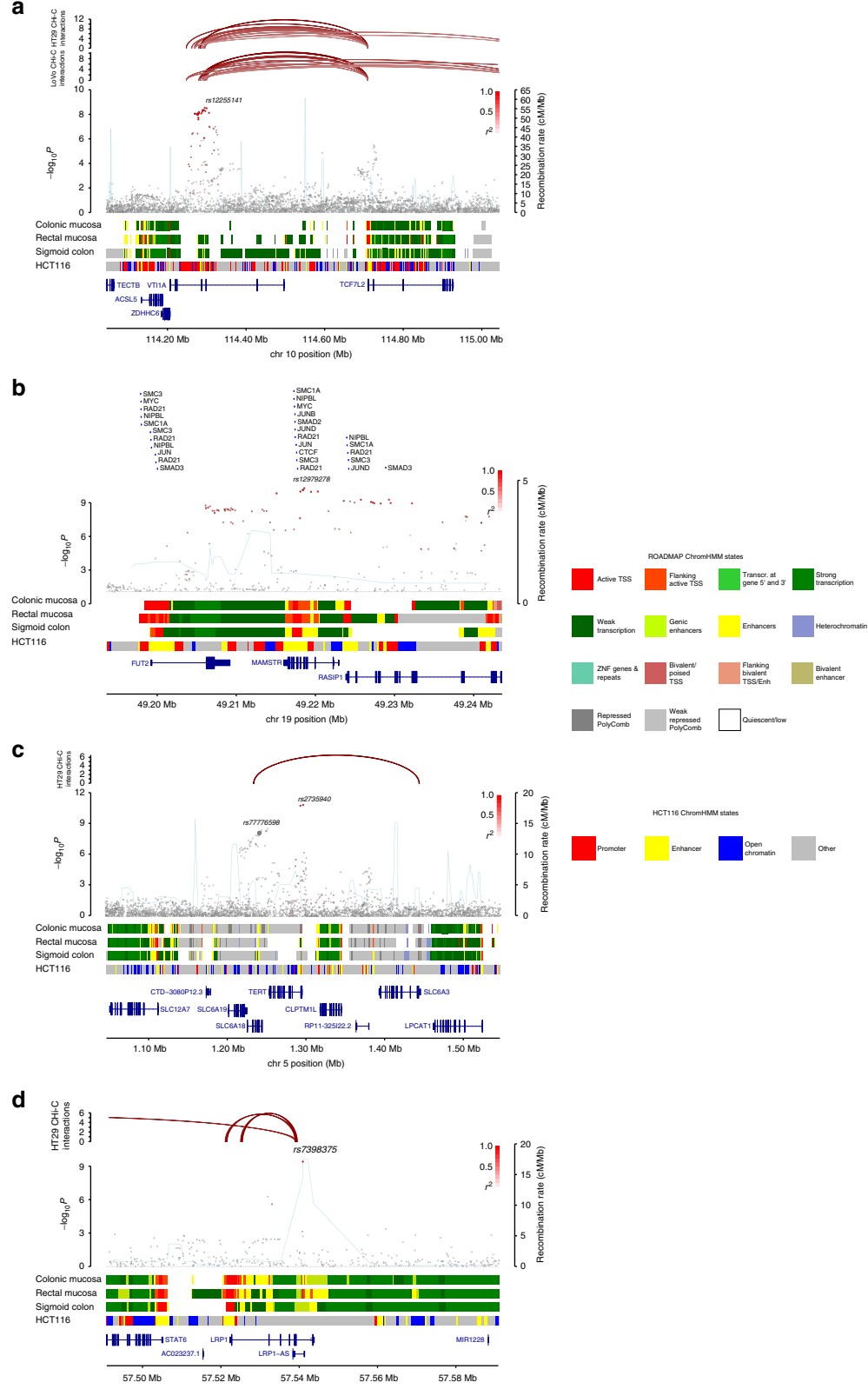

**Fig. 3** Regional plots of exemplar colorectal cancer risk loci. In the main panel, $-\log_{10}P$-values (y-axis) of the SNPs are shown according to their chromosomal positions (x-axis). The colour intensity of each symbol reflects the extent of LD with the top SNP: white ($r^2 = 0$) through to dark red ($r^2 = 1.0$), with $r^2$ estimated from EUR 1000 Genomes data. Genetic recombination rates (cM/Mb) are shown with a light blue line. Physical positions are based on GRCh37 of the human genome. Where available, the upper panel shows Hi-C contacts from HT29 or LoVo. The lower panel shows the chromatin state segmentation track from the Roadmap Epigenomics project (colonic mucosa, rectal mucosa, sigmoid colon), and HCT116. Also shown are the relative positions of genes and transcripts mapping to each region of association. **a** rs12255141 (10q25.2); **b** rs12979278 (19q13); **c** rs2735940 (5p15); **d** rs7398375 (12q13.3)

Project with an additional population matched reference panel: 3882 Sequencing Initiative Suomi (SISu) haplotypes for the FIN study, and 3000 sequenced CRC cases for the DACHS study. We imposed predefined thresholds for imputation quality to retain potential risk variants with MAF >0.5% for validation. Poorly imputed SNPs defined by an information measure <0.80 were excluded. Tests of association between imputed SNPs and CRC were performed under an additive genetic model in SNPTEST v2.5.2[60]. Principal components were added to adjust for population stratification where required (i.e. DACHS, FIN, NSCCG-OncoArray, SCOT and UKBB).

To determine whether specific coding variants within HLA genes contributed to the diverse association signals, we imputed the classical HLA alleles (A, B, C, DQA1, DQB1 and DRB1) and coding variants across the HLA region using SNP2HLA[11]. The imputation was based on a reference panel from the Type 1 Diabetes Genetics Consortium (T1DGC) consisting of genotype data from 5225 individuals of European descent with genotyping data of 8961 common SNPs and indel polymorphisms across the HLA region, and four digit genotyping data of the HLA class I and II molecules. For the X chromosome, genotypes were phased and imputed as for the autosomal chromosome, with the inclusion of the "chrX" flag. X chromosome association analysis was performed in SNPTEST using a maximum likelihood model, assuming complete inactivation of one allele in females and equal effect-size between males and females.

The adequacy of the case-control matching and possibility of differential genotyping of cases and controls was evaluated using a Q–Q plot of test statistics in individual studies (Supplementary Figure 1). Meta-analyses were performed using the fixed-effects inverse-variance method using META v1.7[61]. Cochran's Q-statistic to test for heterogeneity and the $I^2$ statistic to quantify the proportion of the total variation due to heterogeneity were calculated. A Q–Q plot of the meta-analysis test statistics was also performed (Supplementary Figure 1). None of the studies showed evidence of genomic inflation, where $\lambda_{GC}$ values for the CCFR1, CCFR2, COIN, CORSA, Croatia, DACHS, FIN, NSCCG-OncoArray, SCOT, Scotland1, SOCCS/GS, SOCCS/LBC, UKBB, UK1 and VQ58 studies were 1.03, 1.08, 1.09, 1.11, 1.01, 1.01, 1.09, 1.10, 1.08, 1.02, 1.09, 1.04, 1.05, 1.02 and 1.06, respectively. Estimates were calculated using the regression method, as implemented in GenABEL.

**Definition of known and new risk loci.** We sought to identify all associations for CRC previously reported at a significance level $P < 5 \times 10^{-8}$ by referencing the NHGRI-EBI Catalog of published genome-wide association studies, and a literature search for the years 1998–2018 using PubMed (performed January 2018). Additional articles were ascertained through references cited in primary publications. Where multiple studies reported associations in the same region, we only considered the first reported genome-wide significant association. New loci were identified based on SNPs at $P < 5 \times 10^{-8}$ using the meta-analysis summary statistics, with LD correlations from a reference panel of the European 1000 Genomes Project samples combined with UK10K. We only included one SNP per 500 kb interval. To measure the probability of the hits being false positives, the Bayesian False-Discovery Probability (BFDP)[9] was calculated based on a plausible OR of 1.2 (based on the 95th percentile of the meta-analysis OR values) and a prior probability of association of $10^{-5}$. A conditional analysis was performed using Genome-wide Complex Trait Analysis (GCTA)[62], conditioning on the new and known SNPs, and SNPs with $P_{conditioned} < 5 \times 10^{-8}$ and $r^2 > 0.1$ were clumped using PLINK. The NSCCG-Oncoarray data were used to provide the LD reference data.

**Fidelity of imputation.** The reliability of imputation of the novel risk SNPs identified (all with an IMPUTE2 $r^2 > 0.8$) was assessed for 51 SNPs (comprising all new signals not directly genotyped) by examining the concordance between imputed and whole-genome sequenced genotypes in a subset of 201 samples from the CORGI and NSCCG studies. More than 98% concordance was found between the directly sequenced and imputed SNPs (Supplementary Data 14).

**eQTL analysis.** In the INTERMPHEN study, biopsies of normal colorectal mucosa (trios of rectum, proximal colon and distal colon) were obtained from 131 UK individuals with self-reported European ancestry without CRC. Genotyping was performed using the Illumina Infinium Human Core Exome array, with quality control and imputation as above. RNA-seq was performed and data analysed as per the GTEx Project pipeline v7 using the 1000 Genomes and UK10K data as reference. Gene-level expression quantification was based on the GENCODE 19 annotation, collapsed to a single transcript model for each gene using a custom isoform procedure. Gene-level quantification (read counts and TPM values) was performed with RNA-SeQC v1.1.8. Gene expression was normalised using the TMM algorithm, implemented in edgeR, with inverse normal transformation, based on gene expression thresholds of >0.1 Transcripts Per Million (TPM) in ≥20% of samples and ≥6 reads in ≥20% of samples. cis-eQTL mapping was performed separately for proximal colon, distal colon and rectum samples using FastQTL. Principal components for the SNP data and additional covariate factors were identified using Probabilistic Estimation of Expression Residuals (PEER). P-values were generated for each variant-gene pair testing alternative hypothesis that the slope of a linear regression model between genotype and expression deviates from 0. The mapping window was defined as 1 Mb either side of the transcription start site. Beta distribution-adjusted empirical P-values from FastQTL were used to calculate Q-values, and FDR threshold of ≤0.05 was applied to identify genes with a significant eQTL. The normalised effect size of the eQTLs was defined as the slope of the linear regression, and computed as the effect of the alternative allele relative to the reference allele in the human genome reference GRCh37/hg19. MetaTissue was used to generate a "pan-colonic" eQTL measure from the three individual RNA-seq datasets per patient.

To supplement this analysis, we performed SMR analysis[28] including all eQTLs with nominally significant associations ($P < 0.05$). We additionally examined for heterogeneity using the heterogeneity in dependent instruments (HEIDI) test, where $P_{HEIDI} < 0.05$ were considered as reflective of heterogeneity and were excluded.

**Promoter capture Hi-C.** In situ promoter capture Hi-C (CHi-C) on LoVo and HT29 cell lines was performed as previously described[63]. Hi-C and CHi-C libraries were sequenced using HiSeq 2000 (Illumina). Reads were aligned to the GRCh37 build using bowtie2 v2.2.6 and identification of valid di-tags was performed using HiCUP v0.5.9. To declare significant contacts, HiCUP output was processed using CHiCAGO v1.1.8. For each cell line, data from three independent biological replicates were combined to obtain a definitive set of contacts. As advocated, interactions with a score ≥5.0 were considered to be statistically significant[64].

**Chromatin state annotation.** Colorectal cancer risk loci and SNPs in LD ($r^2 > 0.8$) were annotated for putative functional effect based upon ChIP-seq H3K4me1 (C15410194), H3K9me3 (C15410193), H3K27me3 (C15410195) and H3K36me3 (C15410192) for LoVo, and H3K4me1 and H3K9me3 for HT29. ChIP libraries were sequenced using HiSeq 2000 (Illumina) with 100 bp single-ended reads. Generated raw reads were filtered for quality (Phred33 ≥ 30) and length ($n \geq 32$), and adapter sequences were removed using Trimmomatic v0.22. Reads passing filters were then aligned to the human reference (hg19) using BWA v0.6.1. Peak calls are obtained using MACS2 v 2.0.10.07132012.

**Histone mark and transcription factor enrichment analysis.** ChIP-seq data from colon crypt and tumour samples was obtained for H3K27ac and H3K4me1[65]. Multiple samples of the same tissue type or tumour stage were merged together. Additional ChIP-seq data from the Roadmap Epigenomics project[21] was obtained for H3K4me3, H3K27ac, H3K4me1, H3K27me3, H3K9ac, H3K9me3 and H3K36me3 marks in up to 114 tissues. Overlap enrichment analysis of CRC risk SNPs with these peaks was performed using EPIGWAS, as described by Trynka et al.[20]. Briefly, we evaluated if CRC risk SNPs and SNPs in LD ($r^2 > 0.8$) with the sentinel SNP, were enriched at ChIP-seq peaks in tissues by a permutation procedure with $10^5$ iterations.

To examine enrichment in specific TF binding across risk loci, we adapted the variant set enrichment method of Cowper-Sal lari et al.[22]. Briefly, for each risk locus, a region of strong LD (defined as $r^2 > 0.8$ and $D' > 0.8$) was determined, and these SNPs were termed the associated variant set (AVS). ChIP-seq uniform peak data were obtained for LoVo and HT29 cell lines (198 and 29 experiments, respectively)[66] and the above described histone marks. For each of these marks, the overlap of the SNPs in the AVS and the binding sites was determined to produce a mapping tally. A null distribution was produced by randomly selecting SNPs with the same characteristics as the risk-associated SNPs, and the null mapping tally calculated. This process was repeated $10^5$ times, and P-values calculated as the proportion of permutations where the null mapping tally was greater or equal to the AVS mapping tally. An enrichment score was calculated by normalising the tallies to the median of the null distribution. Thus, the enrichment score is the number of standard deviations of the AVS mapping tally from the median of the null distribution tallies.

**Functional annotation.** For the integrated functional annotation of risk loci, LD blocks were defined as all SNPs in $r^2 > 0.8$ with the sentinel SNP. Risk loci were then annotated with five types of functional data: (i) presence of a CHi-C contact linking to a gene promoter, (ii) presence of an association from eQTL, (iii) presence of a regulatory state, (iv) evidence of TF binding, and (v) presence of a non-synonymous coding change. Candidate causal genes were then assigned to CRC risk loci using the target genes implicated in annotation tracks (i), (ii), (iiii) and (iv). If the data supported multiple gene candidates, the gene with the highest number of individual functional data points was considered as the candidate. Where multiple genes had the same number of data points, all genes were listed. Direct nonsynonymous coding variants were allocated additional weighting. Competing mechanisms for the same gene (e.g. both coding and promoter variants) were allowed for. Finally, if no evidence was provided by these criteria, if the lead SNP was intronic we assigned candidacy on this basis, or if intergenic the nearest gene neighbour. Chromatin data were obtained from HaploReg v4 and regulatory regions from Ensembl.

Regional plots were created using visPIG[67], using the data described above. We used ChromHMM to integrate DNAse, H3K4me3, H3K4me1, H3K27ac, Pol2 and CTCF states from the CRC cell line HCT116 using a multivariate Hidden Markov Model[68]. Chromatin annotation tracks for colonic mucosa (E075), rectal mucosa (E101) and sigmoid colon (E106) were obtained from the Roadmap Epigenomics

project[21], using the core 15-state model data based on H3K4me3, H3K4me1, H3K36me3, H3K27me3 and H3K9me3 marks.

**Transcription factor binding disruption analysis.** To determine if the risk variants or their proxies were disrupting motif binding sites, we used the motifbreakR package[69]. This tool predicts the effects of variants on TF binding motifs, using position probability matrices to determine the likelihood of observing a particular nucleotide at a specific position within a TF binding site. We tested the SNPs by estimating their effects on over 2,800 binding motifs as characterised by ENCODE, FactorBook, HOCOMOCO and HOMER. Scores were calculated using the relative entropy algorithm.

**Heritability analysis.** We used LDAK[35] to estimate the polygenic variance (i.e. heritability) ascribable to SNPs from summary statistic data for the GWAS datasets which were based on unselected cases (i.e. CORSA, COIN, Croatia, DACHS, FIN, SCOT, Scotland1, SOCCS/GS, SOCCS/LBC, UKBB and VQ58). SNP-specific expected heritability, adjusted for LD, MAF and genotype certainty, was calculated from the UK10K and 1000 Genomes data. Individuals were excluded if they were closely related, had divergent ancestry from CEU, or had a call rate <0.99. SNPs were excluded if they showed deviation from HWE with $P < 1 \times 10^{-5}$, genotype yield <95%, MAF <1%, SNP imputation score <0.99, and the absence of the SNP in the GWAS summary statistic data. This resulted in a total 6,024,731 SNPs used to estimate the heritability of CRC.

To estimate the sample size required to detect a given proportion of the GWAS heritability we implemented a likelihood-based approach to model the effect-size distribution[36], using association statistics from the meta-analysis, and LD information from individuals of European ancestry in the 1000 Genomes Project Phase 3. LD values were based on an $r^2$ threshold of 0.1 and a window size of 1MB. The goodness of fit of the observed distribution of $P$-values against the expected from a two-component model (single normal distribution) and a three-component model (mixture of two normal distributions) were assessed, and a better fit was observed for the latter model. The percentage of GWAS heritability explained for a projected sample size was determined using this model, based on power calculations for the discovery of genome-wide significant SNPs. The genetic variance explained was calculated as the proportion of total GWAS heritability explained by SNPs reaching genome-wide significance at a given sample size. The 95% confidence intervals were determined using $10^5$ simulations.

**Cross-trait genetic correlation.** LD score regression[39] was used to determine if any traits were correlated with CRC risk. GWAS summary data were obtained for allergy, asthma, coronary artery disease, fatty acids, lipids (total cholesterol, high density lipoprotein, low-density lipoprotein, triglycerides), auto-immune diseases (Crohn's disease, rheumatoid arthritis, atopic dermatitis, celiac disease, multiple sclerosis, primary biliary cirrhosis, inflammatory bowel disease, ulcerative colitis, systemic lupus erythematosus), anthropometric measures (BMI, height, body fat), glucose sensitivity (fasting glucose, fasting insulin, HbA1c), childhood measures (birth weight, birth length, childhood obesity, childhood BMI), eGFR and type 2 diabetes. All data were obtained for European populations. Summary statistics were reformatted to be consistent, and constrained to HapMap3 SNPs as these have been found to generally impute well. LD Scores were determined using 1000 Genomes European data.

**Familial risk explained by risk SNPs.** Under a multiplicative model, the contribution of risk SNPs to the familial risk of CRC was calculated from $\sum_k \frac{\log \lambda_k}{\log \lambda_0}$, where $\lambda_0$ is the familial risk to first-degree relatives of CRC cases, assumed to be 2.2[38], and $\lambda_k$ is the familial relative risk associated with SNP $k$, calculated as $\lambda_k = \frac{p_k r_k^2 + q_k}{(p_k r_k + q_k)^2}$, where $p_k$ is the risk allele frequency for SNP $k$, $q_k = 1 - p_k$, and $r_k$ is the estimated per-allele OR from the meta-analysis[70]. The OR estimates were adjusted for the winner's curse using the FDR Inverse Quantile Transformation (FIQT) method[37]. We constructed a PRS including all 79 CRC risk SNPs discovered or validated by this GWAS in the risk-score modelling. The distribution of risk on an RR scale in the population is assumed to be log-normal with arbitrary population mean $\mu$ set to $-\sigma^2/2$ and variance $\sigma^2 = 2 \sum_k p_k (1 - p_k) \beta^2$ where $\beta$ and $p$ correspond to the log odds ratio and the risk allele frequency, respectively, for SNP $k$. The distribution of PRS among cases is right-shifted by $\sigma^2$ so that the overall mean PRS is 1.0[71]. The risk distribution was also performed assuming all common variation, using $\sigma^2 = \log(\lambda_{sib}^2)$, where $\lambda_{sib} = 1.79$, as determined using the heritability estimate from GCTA.

**Pathway analysis.** SNPs were assigned to genes as described in the functional annotation section. The genes that mapped to genome-wide significant CRC risk SNPs were analysed using InBio Map, a manually curated database of protein-protein interactions.

Gene set enrichment was calculated using GenGen. Enrichment scores were calculated using the meta-analysis results and were based on $10^3$ permutations on the $\chi^2$ values between SNPs. Pathway definitions were obtained from the Bader Lab[33], University of Toronto, July 2018 release. This data contained pathway information from Gene Ontology (GO), Reactome, HumanCyc, MSigdb C2 (curated dataset), NCI Pathway, NetPath and PANTHER for a total of 7269

pathways. GO annotations that were inferred computationally were excluded. To avoid biasing the results, the meta-analysis SNPs were pruned to only those with an $r^2 < 0.1$ and a distance greater than 500 kb. Pathways were visualised using Cytoscape v3.6.1, together with the EnrichmentMap v3.1.0 and AutoAnnotate v1.2 plugins. Only pathways with an FDR <0.05 and edges with a similarity coefficient (number of shared genes between pathways) >0.55 were displayed.

**URLs.** Bader Lab pathway data: http://download.baderlab.org/EM_Genesets/July_01_2018/Human/symbol/
 FastQTL: https://github.com/francois-a/fastqtl
 GTEx: https://www.gtexportal.org/home/
 InBioMap: https://www.intomics.com/inbio/map/#home
 LD scores: https://data.broadinstitute.org/alkesgroup/LDSCORE/
 NHGRI-EBI GWAS Catalog: https://www.ebi.ac.uk/gwas/
 PredictDB: http://predictdb.org/
 Roadmap Epigenomics data: https://egg2.wustl.edu/roadmap/web_portal/chr_state_learning.html
 SYSCOL: http://syscol-project.eu/
 UK Biobank: http://www.ukbiobank.ac.uk/scientists-3/genetic-data/

**Reporting summary.** Further information on research design is available in the Nature Research Reporting Summary linked to this article.

## Data availability
The SCOT data can be requested through the TransSCOT committee according to the ethical permissions obtained as part of the clinical trial approval. The PRACTICAL and BCAC consortium control data are available through the respective Data Access Coordination Committees (http://practical.icr.ac.uk and http://bcac.ccge.medschl.cam.ac.uk/) and the Heinz Nixdorf Recall Study control data can be requested through https://www.uni-due.de/recall-studie/die-studien/hnr/. UK Biobank data can be obtained through http://www.ukbiobank.ac.uk/. The Colon Cancer Family Registry data can be obtained through http://coloncfr.org/.
Finnish cohort samples can be requested from THL Biobank https://thl.fi/en/web/thl-biobank. Hi-C, CHi-C, and histone ChIPseq sequencing data have been deposited in the European Genome-phenome Archive (EGA) under the accession code EGAS00001001946. The remaining data are contained within the Supplementary Files or available from the authors upon reasonable request.

## Code availability
All bioinformatics and statistical analysis tools used are open source.

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

## Acknowledgements

At the Institute of Cancer Research, this work was supported by Cancer Research UK (C1298/A25514). Additional support was provided by the National Cancer Research Network. In Edinburgh, the work was supported by Programme Grant funding from Cancer Research UK (C348/A12076) and by funding for the infrastructure and staffing of

the Edinburgh CRUK Cancer Research Centre. In Birmingham, funding was provided by Cancer Research UK (C6199/A16459). We are grateful to many colleagues within UK Clinical Genetics Departments (for CORGI) and to many collaborators who participated in the VICTOR, QUASAR2 and SCOT trials.

We also thank colleagues from the UK National Cancer Research Network (for NSCCG). Support from the European Union [FP7/207–2013, grant 258236] and FP7 collaborative project SYSCOL and COST Action in the UK is also acknowledged [BM1206]. The COIN and COIN-B trials were funded by Cancer Research UK and the Medical Research Council and were conducted with the support of the National Institute of Health Research Cancer Research Network. COIN and COIN-B translational studies were supported by the Bobby Moore Fund from Cancer Research UK, Tenovus, the Kidani Trust, Cancer Research Wales and the National Institute for Social Care and Health Research Cancer Genetics Biomedical Research Unit (2011–2014).

We thank the High-Throughput Genomics Group at the Wellcome Trust Centre for Human Genetics (funded by Wellcome Trust grant reference 090532/Z/09/Z) and the Edinburgh Clinical Research Facility (ECRF) Genetics Core, Western General Hospital, Edinburgh, for the generation of genotyping data.

We thank the Lothian Birth Cohorts' members, investigators, research associates, and other team members. We thank the Edinburgh Clinical Research Facility (ECRF) Genetics Core, Western General Hospital, Edinburgh, for genotyping. Lothian Birth Cohorts' data collection is supported by the Disconnected Mind project (funded by Age UK), and the Biotechnology and Biological Sciences Research Council (BBSRC, for genotyping; BB/F019394/1) and undertaken within the University of Edinburgh Centre for Cognitive Ageing and Cognitive Epidemiology (funded by the BBSRC and Medical Research Council RC as part of the LLHW [MR/K026992/1]). ET was supported by Cancer Research UK CDF (C31250/A22804). This research has been conducted using the UK Biobank Resource under Application Number 7441.

Generation Scotland received core support from the Chief Scientist Office of the Scottish Government Health Directorates [CZD/16/6] and the Scottish Funding Council [HR03006]. Genotyping of the GS:SFHS samples was carried out by the Genetics Core Laboratory at the Clinical Research Facility, University of Edinburgh and was funded by the Medical Research Council UK and the Wellcome Trust (Wellcome Trust Strategic Award "STratifying Resilience and Depression Longitudinally" (STRADL) [104036/Z/14/Z]).

CFR was supported by a Marie Sklodowska-Curie Intra-European Fellowship Action and received considerable help from many staff in the Department of Endoscopy at the John Radcliffe Hospital in Oxford.

In Finland, this work was supported by grants from the Academy of Finland [Finnish Center of Excellence Program 2012–2017, 250345 and 2018–2025, 312041], the Jane and Aatos Erkko Foundation, the Finnish Cancer Society [personal grant to K.P.], the European Research Council [ERC; 268648], the Sigrid Juselius Foundation, SYSCOL, the Nordic Information for Action eScience Center (NIASC), the Nordic Center of Excellence financed by NordForsk [project 62721, personal grant to K.P.] and State Research Funding of Kuopio University Hospital [B1401]. We acknowledge the computational resources provided by the ELIXIR node, hosted at the CSC–IT Center for Science, Finland, and funded by the Academy of Finland [grants 271642 and 263164], the Ministry of Education and Culture, Finland. V.S. was supported by the Finnish Academy [grant number 139635] and the Finnish Foundation for Cardiovascular Research. J.-P.M. was funded by The Finnish Cancer Foundation and The Jane and Aatos Erkko Foundation. Sample collection and genotyping in the Finnish Twin Cohort has been supported by the Wellcome Trust Sanger Institute, ENGAGE—European Network for Genetic and Genomic Epidemiology, FP7-HEALTH-F4–2007; [grant agreement number 201413], the National Institute of Alcohol Abuse and Alcoholism [grants AA-12502 and AA-00145; to R.J.R. and K02AA018755 to D.M.D.] and the Academy of Finland [grants 100499, 205585, 265240 and 263278 to J.K.].

The work of the Colon Cancer Family Registry (CCFR) was supported by the National Cancer Institute (NCI) of the National Institutes of Health (NIH) under Award number U01 CA167551. The CCFR Illumina GWAS was supported by the NCI/NIH under Award Numbers U01 CA122839 and R01 CA143237 to G.C. The content of this manuscript does not necessarily reflect the views or policies of the NCI or any of the collaborating centres in the CCFR, nor does mention of trade names, commercial products, or organizations imply endorsement by the US Government or the CCFR.

The CORSA study was funded by FFG BRIDGE (grant 829675, to A.G.), the "Herzfelder'sche Familienstiftung" (grant to A.G.) and was supported by COST Action BM1206. We kindly thank all individuals who agreed to participate in the CORSA study. Furthermore, we thank all cooperating physicians and students and the Biobank Graz of the Medical University of Graz.

The DACHS study was supported by grants from the German Research Council (Deutsche Forschungsgemeinschaft, BR 1704/6–1, BR 1704/6–3, BR 1704/6–4, BR 1704/6–6 and CH 117/1–1), and the German Federal Ministry of Education and Research (01KH0404, 01ER0814, 01ER0815 and 01ER1505A, 01ER1505B). We thank all participants and cooperating clinicians, and Ute Handte-Daub, Ansgar Brandhorst, Muhabbet Celik and Ursula Eilber for excellent technical assistance.

The Croatian study was supported through the 10,001 Dalmatians Project, and institutional support of University Hospital for Tumours, Sestre milosrdnice University Hospital Center.

James East and Simon Leedham were funded by the National Institute for Health Research (NIHR) Oxford Biomedical Research Centre (BRC). The views expressed not necessarily those of the NHS, the NIHR or the Department of Health.

We acknowledge use of genotype data from the British 1958 Birth Cohort DNA collection, which was funded by the Medical Research Council Grant G0000934 and the Wellcome Trust Grant 068545/Z/02. A full list of the investigators who contributed to the generation of the data is available from http://www.wtccc.org.uk.

The BCAC study would not have been possible without the contributions of the following: Manjeet K. Bolla, Qin Wang, Kyriaki Michailidou and Joe Dennis. BCAC is funded by Cancer Research UK (C1287/A10118, C1287/A16563). For the BBCS study, we thank Eileen Williams, Elaine Ryder-Mills, Kara Sargus. The BBCS is funded by Cancer Research UK and Breast Cancer Now and acknowledges NHS funding to the National Institute of Health Research (NIHR) Biomedical Research Centre (BRC) and the National Cancer Research Network (NCRN). We thank the participants and the investigators of EPIC (European Prospective Investigation into Cancer and Nutrition). The coordination of EPIC is financially supported by the European Commission (DG-SANCO) and the International Agency for Research on Cancer. The national cohorts are supported by: Ligue Contre le Cancer, Institut Gustave Roussy, Mutuelle Générale de l'Education Nationale, Institut National de la Santé et de la Recherche Médicale (INSERM) (France); German Cancer Aid, German Cancer Research Center (DKFZ), Federal Ministry of Education and Research (BMBF) (Germany); the Hellenic Health Foundation, the Stavros Niarchos Foundation (Greece); Associazione Italiana per la Ricerca sul Cancro-AIRC-Italy and National Research Council (Italy); Dutch Ministry of Public Health, Welfare and Sports (VWS), Netherlands Cancer Registry (NKR), LK Research Funds, Dutch Prevention Funds, Dutch ZON (Zorg Onderzoek Nederland), World Cancer Research Fund (WCRF), Statistics Netherlands (The Netherlands); Health Research Fund (FIS), PI13/00061 to Granada, PI13/01162 to EPIC-Murcia, Regional Governments of Andalucía, Asturias, Basque Country, Murcia and Navarra, ISCIII RETIC (RD06/0020) (Spain); Cancer Research UK (14136 to EPIC-Norfolk; C570/A16491 and C8221/A19170 to EPIC-Oxford), Medical Research Council (1000143 to EPIC-Norfolk, MR/M012190/1 to EPIC-Oxford) (United Kingdom). We thank the SEARCH and EPIC teams, which were funded by a programme grant from Cancer Research UK (C490/A10124) and supported by the UK NIHR BRC at the University of Cambridge. We thank Breast Cancer Now and the Institute of Cancer Research (ICR) for support and funding of the UKBGS, and the study participants, study staff, and the doctors, nurses and other health-care providers and health information sources who have contributed to the study.

Genotyping of the PRACTICAL consortium OncoArray was funded by the US National Institutes of Health (NIH) [U19 CA 148537 for ELucidating Loci Involved in Prostate cancer SuscEptibility (ELLIPSE) project and X01HG007492 to the Center for Inherited Disease Research (CIDR) under contract number HHSN268201200008I]. Additional analytic support was provided by NIH NCI U01 CA188392 (PI: Schumacher). The PRACTICAL consortium was supported by Cancer Research UK Grants C5047/A7357, C1287/A10118, C1287/A16563, C5047/A3354, C5047/A10692, C16913/A6135, European Commission's Seventh Framework Programme grant agreement n° 223175 (HEALTH-F2–2009–223175), and The National Institute of Health (NIH) Cancer Post-Cancer GWAS initiative grant: No. 1 U19 CA 148537–01 (the GAME-ON initiative). We would also like to thank the following for funding support: The Institute of Cancer Research and The Everyman Campaign, The Prostate Cancer Research Foundation, Prostate Research Campaign UK (now Prostate Action), The Orchid Cancer Appeal, The National Cancer Research Network UK, The National Cancer Research Institute (NCRI) UK. We are grateful for support of NIHR funding to the NIHR Biomedical Research Centre at The Institute of Cancer Research and The Royal Marsden NHS Foundation Trust, the Spanish Instituto de Salud Carlos III (ISCIII) an initiative of the Spanish Ministry of Economy and Innovation (Spain), and the Xunta de Galicia (Spain).

The APBC BioResource, which forms part of the PRACTICAL consortium, consists of the following members: Wayne Tilley, Gail Risbridger, Renea Taylor, Lisa Horvath, Vanessa Hayes, Lisa Butler, Trina Yeadon, Allison Eckert, Anne-Maree Haynes, Melissa Papargiris.

We are grateful for the provision of public data from the GTEx consortium.

Finally, the authors gratefully acknowledge the participation of patients, their families, and controls in the relevant studies.

## Author contributions

Study concept and design: R.S.H., I.T. and M.G.D. Patient recruitment: S.P. and L.M. Sample preparation and genotyping: A.Holroyd., P.B. Primary data analysis: P.J.L., M.T., C.F.-R. and J.F.-T. Additional analysis: J.Studd., G.O., A.Sud., S.F., V.S., C.P., S.Briggs., L.M., E.Jaeger., A.S.-O., J.E. and S.L. Provided sample data: E.T., P.V.-S., L.Z., A.C., H.C., C.H., S.H., I.J.D., J.Starr., R.A., E.Johnstone., H.W., L.G., M.P., D.K., R.Kerr., T.M., R.Kaplan., N.A.-T., J.P.C., K.P., L.A.A., U.A.H., T.C., T.T., J.K., E.K., A.-P.S., J.G.E., H.R., P.K., E.P., P.J., V.S., S.R., A.P., L.R.-S., A.L., J.B., J.-P.M., D.D.B., A.-K.W., J.H., M.E.J., N.M.L., P.A.N., S.G., D.D., G.C., P.Hoffmann., M.M.N., K.-H.J., D.F.E., P.D.P.P., J.Peto., F.C., A.Swerdlow., R.A.E., Z.K.-J., K.M., N.P., PRACTICAL Consortium, A.Harkin., K.A., J.M., J.Paul., T.I., M.S., K.B., J.C.-C., M.H., H.B., I.K., P.M., P.Hofer., S.Brezina. and A.G. Writing manuscript: R.S.H., I.T., M.G.D and P.J.L. All authors read and approved the final version of the manuscript.

## Additional information

**Competing interests:** D.K. is a founder and shareholder of Oxford Cancer Biomarkers. V.S. has participated in a conference trip sponsored by Novo Nordisk and received an honorarium from the same source for participating in an advisory board meeting. The remaining authors declare no competing interests.

Philip J. Law[1], Maria Timofeeva[2], Ceres Fernandez-Rozadilla[3,4], Peter Broderick[1], James Studd[1], Juan Fernandez-Tajes[5], Susan Farrington[2], Victoria Svinti[2], Claire Palles[6], Giulia Orlando[1], Amit Sud[1], Amy Holroyd[1], Steven Penegar[1], Evropi Theodoratou[2,7], Peter Vaughan-Shaw[2], Harry Campbell[2,7], Lina Zgaga[2,8], Caroline Hayward[9], Archie Campbell[10], Sarah Harris[10,11,12], Ian J. Deary[10,11], John Starr[10,13,14], Laura Gatcombe[4], Maria Pinna[4], Sarah Briggs[4], Lynn Martin[4], Emma Jaeger[4], Archana Sharma-Oates[4], James East[15], Simon Leedham[5,14], Roland Arnold[16], Elaine Johnstone[17], Haitao Wang[17], David Kerr[18], Rachel Kerr[17], Tim Maughan[17], Richard Kaplan[19], Nada Al-Tassan[20], Kimmo Palin[21], Ulrika A. Hänninen[21], Tatiana Cajuso[21], Tomas Tanskanen[21], Johanna Kondelin[21], Eevi Kaasinen[21], Antti-Pekka Sarin[22], Johan G. Eriksson[23,24], Harri Rissanen[25], Paul Knekt[25], Eero Pukkala[26,27], Pekka Jousilahti[25], Veikko Salomaa[25], Samuli Ripatti[22,28,29], Aarno Palotie[22,30], Laura Renkonen-Sinisalo[31], Anna Lepistö[31], Jan Böhm[32], Jukka-Pekka Mecklin[33,34], Daniel D. Buchanan[35,36,37], Aung-Ko Win[38], John Hopper[38], Mark E. Jenkins[38], Noralane M. Lindor[39], Polly A. Newcomb[40], Steven Gallinger[41], David Duggan[42], Graham Casey[43], Per Hoffmann[44,45], Markus M. Nöthen[45,46], Karl-Heinz Jöckel[47], Douglas F. Easton[48,49], Paul D.P. Pharoah[48,49], Julian Peto[50], Federico Canzian[51], Anthony Swerdlow[1,52], Rosalind A. Eeles[1,53], Zsofia Kote-Jarai[1], Kenneth Muir[54,55], Nora Pashayan[56,57], The PRACTICAL consortium, Andrea Harkin[58], Karen Allan[58], John McQueen[58], James Paul[58], Timothy Iveson[59], Mark Saunders[60], Katja Butterbach[61], Jenny Chang-Claude[62,63], Michael Hoffmeister[61], Hermann Brenner[61,64,65], Iva Kirac[66], Petar Matošević[67], Philipp Hofer[68], Stefanie Brezina[68], Andrea Gsur[68], Jeremy P. Cheadle[69], Lauri A. Aaltonen[21], Ian Tomlinson[4], Richard S. Houlston[1] & Malcolm G. Dunlop[2]

[1]Division of Genetics and Epidemiology, The Institute of Cancer Research, London SW7 3RP, UK. [2]Colon Cancer Genetics Group, Medical Research Council Human Genetics Unit, Institute of Genetics and Molecular Medicine, Western General Hospital, University of Edinburgh, Edinburgh EH4 2XU, UK. [3]Grupo de Medicina Xenómica, Fundación Pública Galega de Medicina Xenómica, Instituto de Investigación de Santiago, Santiago de Compostela 15706, Spain. [4]Cancer Genetics and Evolution Laboratory, Institute of Cancer and Genomic Sciences, University of Birmingham, Vincent Drive, Edgbaston, Birmingham B15 2TT, UK. [5]Wellcome Centre for Human Genetics, McCarthy Group, Roosevelt Drive, Oxford OX3 7BN, UK. [6]Gastrointestinal Cancer Genetics Laboratory, Institute of Cancer and Genomic Sciences, University of Birmingham, Vincent Drive, Edgbaston, Birmingham B15 2TT, UK. [7]Centre for Global Health Research, Usher Institute, University of Edinburgh, Edinburgh EH8 9AG, UK. [8]Department of Public Health and Primary Care, Institute of Population Health, Trinity College Dublin, University of Dublin, Dublin D02 PN40, Ireland. [9]Medical Research Council Human Genetics Unit, Institute of Genetics and Molecular Medicine, Western General Hospital, University of Edinburgh, Edinburgh EH4 2XU, UK. [10]Generation Scotland, Centre for Genomic and Experimental Medicine, MRC Institute of Genetics and Molecular Medicine, University of Edinburgh, Edinburgh EH4 2XU, UK. [11]Centre for Cognitive Ageing and Cognitive Epidemiology, University of Edinburgh, Edinburgh EH8 9JZ, UK. [12]Department of Psychology, University of Edinburgh, Edinburgh EH8 9JZ, UK. [13]Medical Genetics Section, Centre for Genomics and Experimental Medicine, Institute of Genetics and Molecular Medicine, University of Edinburgh, Edinburgh EH4 2XU, UK. [14]Alzheimer Scotland Dementia Research Centre, University of Edinburgh, Edinburgh EH8 9JZ, UK. [15]Translational Gastroenterology Unit, Nuffield Department. of Medicine, University of Oxford, John Radcliffe Hospital, Oxford OX3 9DU, UK. [16]Cancer Bioinfomatics Laboratory, Institute of Cancer and Genomic Sciences, University of Birmingham, Vincent Drive, Edgbaston, Birmingham B15 2TT, UK. [17]Department of Oncology, University of Oxford, Old Road Campus Research Building, Oxford OX3 7LE, UK. [18]Nuffield Department of Clinical Laboratory Sciences, John Radcliffe Hospital, University of Oxford, Oxford OX3 9DU, UK. [19]Medical Research Council Clinical Trials Unit, Aviation House, 125 Kingsway, London WC2B 6NH, UK. [20]Department of Genetics, King Faisal Specialist Hospital and Research Center, Riyadh 11211, Saudi Arabia. [21]Department of Medical and Clinical Genetics, Medicum and Genome-Scale Biology Research Program, Research Programs Unit, University of Helsinki, Helsinki 00014, Finland.

[22]Institute for Molecular Medicine Finland (FIMM), University of Helsinki, Helsinki 00014, Finland. [23]Folkhälsan Research Centre, 00250 Helsinki, Finland. [24]Unit of General Practice and Primary Health Care, University of Helsinki and Helsinki University Hospital, Helsinki 00014, Finland. [25]National Institute for Health and Welfare, Helsinki 00271, Finland. [26]Finnish Cancer Registry, Institute for Statistical and Epidemiological Cancer Research, Helsinki, Finland, and Faculty of Social Sciences, University of Tampere, Tampere 33014, Finland. [27]Faculty of Social Sciences, University of Tampere, Tampere 33014, Finland. [28]Department of Public Health, University of Helsinki, Helsinki 00014, Finland. [29]Broad Institute of MIT and Harvard, Cambridge, MA 02142, USA. [30]Analytic and Translational Genetics Unit, Department of Medicine, Massachusetts General Hospital, Boston, MA 02114, USA. [31]Department of Surgery, Abdominal Center, Helsinki University Hospital, Helsinki 00029, Finland. [32]Department of Pathology, Central Finland Central Hospital, Jyväskylä 40620, Finland. [33]Department of Surgery, Jyväskylä Central Hospital, Jyväskylä 40620, Finland. [34]Department of Health Sciences, Faculty of Sport and Health Sciences, University of Jyväskylä, Jyväskylä 40014, Finland. [35]Colorectal Oncogenomics Group, Department of Clinical Pathology, The University of Melbourne, Parkville, Victoria 3010, Australia. [36]Victorian Comprehensive Cancer Centre, University of Melbourne, Centre for Cancer Research, Parkville, Victoria 3010, Australia. [37]Genomic Medicine and Family Cancer Clinic, Royal Melbourne Hospital, Parkville, VIC 3010, Australia. [38]Centre for Epidemiology and Biostatistics, The University of Melbourne, Melbourne, VIC 3010, Australia. [39]Department of Health Sciences Research, Mayo Clinic, Scottsdale, AZ 85259, USA. [40]Cancer Prevention Program, Fred Hutchinson Cancer Research Center, Seattle, WA 98109, USA. [41]Mount Sinai Hospital, Lunenfeld-Tanenbaum Research Institute, Toronto ON M5G 1X5, Canada. [42]Translational Genomics Research Institute (TGen), An Affiliate of City of Hope, Phoenix, AZ 85004, USA. [43]Center for Public Health Genomics, University of Virginia, Virginia, VA 22903, USA. [44]Human Genomics Research Group, Department of Biomedicine, University of Basel, Basel 4031, Switzerland. [45]Department of Genomics, Life & Brain Center, University of Bonn, Bonn 53127, Germany. [46]Institute of Human Genetics, University of Bonn School of Medicine & University Hospital Bonn, Bonn 53127, Germany. [47]Institute for Medical Informatics, Biometry and Epidemiology, University Hospital Essen, University of Duisburg-Essen, Essen 45147, Germany. [48]Centre for Cancer Genetic Epidemiology, Department of Oncology, University of Cambridge, Cambridge CB1 8RN, UK. [49]Centre for Cancer Genetic Epidemiology, Department of Public Health and Primary Care, University of Cambridge, Cambridge CB1 8RN, UK. [50]Department of Non-Communicable Disease Epidemiology, London School of Hygiene and Tropical Medicine, London WC1E 7HT, UK. [51]Genomic Epidemiology Group, German Cancer Research Center (DKFZ), Heidelberg 69120, Germany. [52]Division of Breast Cancer Research, The Institute of Cancer Research, London SW3 6JB, UK. [53]Royal Marsden NHS Foundation Trust, London SW3 6JJ, UK. [54]Division of Population Health, Health Services Research and Primary Care, University of Manchester, Manchester M13 9PL, UK. [55]Warwick Medical School, University of Warwick, Coventry CV4 7HL, UK. [56]Department of Applied Health Research, University College London, London WC1E 7HB, UK. [57]Centre for Cancer Genetic Epidemiology, Department of Oncology, Strangeways Laboratory, University of Cambridge, Cambridge CB1 8RN, UK. [58]Cancer Research UK Clinical Trials Unit, Institute of Cancer Sciences, University of Glasgow, Glasgow G61 1BD, UK. [59]University Hospital Southampton NHS Foundation Trust, Southampton SO16 6YD, UK. [60]The Christie NHS Foundation Trust, Manchester M20 4BX, UK. [61]Division of Clinical Epidemiology and Aging Research, Deutsches Krebsforschungszentrum, 69120 Heidelberg, Germany. [62]Unit of Genetic Epidemiology, German Cancer Research Center (DKFZ), Heidelberg 69120, Germany. [63]University Cancer Center Hamburg, University Medical Center Hamburg-Eppendorf, Hamburg 20251, Germany. [64]German Cancer Consortium (DKTK), German Cancer Research Center (DKFZ), Heidelberg 69120, Germany. [65]Division of Preventive Oncology, German Cancer Research Center (DKFZ) and National Center for Tumor Diseases (NCT), Heidelberg 69120, Germany. [66]Department of Surgical Oncology, University Hospital for Tumours, Sestre milosrdnice University Hospital Centre, Zagreb 10000, Croatia. [67]Department of Surgery, University Hospital Center Zagreb, 10000 Zagreb, Croatia. [68]Department of Medicine I, Institute of Cancer Research, Medical University of Vienna, Borschkegasse 8a, 1090 Vienna, Austria. [69]Division of Cancer and Genetics, School of Medicine, Cardiff University, Cardiff CF14 4XN, UK. These authors contributed equally: Philip J. Law, Maria Timofeeva, Ceres Fernandez-Rozadilla. These authors jointly supervised this work: Ian Tomlinson, Richard S. Houlston, Malcolm G. Dunlop. Additional members from the Prostate Cancer Association Group to Investigate Cancer Associated Alterations in the Genome (PRACTICAL) consortium are listed below.

## The PRACTICAL consortium

Brian E. Henderson[70], Christopher A. Haiman[70], Fredrick R. Schumacher[71,72], Ali Amin Al Olama[57,73], Sara Benlloch[1,57], Sonja I. Berndt[74], David V. Conti[70], Fredrik Wiklund[75], Stephen Chanock[70], Susan Gapstur[76], Victoria L. Stevens[76], Catherine M. Tangen[77], Jyotsna Batra[78,79], Judith Clements[78,79], Henrik Gronberg[75], Johanna Schleutker[80,81], Demetrius Albanes[74], Alicja Wolk[82,83], Catharine West[84], Lorelei Mucci[85], Géraldine Cancel-Tassin[86,87], Stella Koutros[74], Karina Dalsgaard Sorensen[88,89], Eli Marie Grindedal[90], David E. Neal[91,92,93,94], Freddie C. Hamdy[93,94], Jenny L. Donovan[95], Ruth C. Travis[96], Robert J. Hamilton[97], Sue Ann Ingles[70], Barry S. Rosenstein[98,99], Yong-Jie Lu[100], Graham G. Giles[101,102], Adam S. Kibel[103], Ana Vega[104], Manolis Kogevinas[105,106,107,108], Kathryn L. Penney[109], Jong Y. Park[110], Janet L. Stanford[111,112], Cezary Cybulski[113], Børge G. Nordestgaard[114,115], Christiane Maier[116], Jeri Kim[117], Esther M. John[118,119], Manuel R. Teixeira[120,121], Susan L. Neuhausen[122], Kim De Ruyck[123], Azad Razack[124], Lisa F. Newcomb[111,125], Marija Gamulin[126], Radka Kaneva[127], Nawaid Usmani[128,129], Frank Claessens[130], Paul A. Townsend[131], Manuela Gago-Dominguez[3,132], Monique J. Roobol[133], Florence Menegaux[134], Kay-Tee Khaw[135], Lisa Cannon-Albright[136,137], Hardev Pandha[138] & Stephen N. Thibodeau[139]

[70]Department of Preventive Medicine, Keck School of Medicine, University of Southern California/Norris Comprehensive Cancer Center, Los Angeles, CA 90033, USA. [71]Department of Epidemiology and Biostatistics, Case Western Reserve University, Cleveland, OH 44106, USA. [72]Seidman Cancer Center, University Hospitals, Cleveland, OH 44106, USA. [73]Department of Clinical Neurosciences, University of Cambridge, Cambridge CB2 1TN, UK. [74]Division of Cancer Epidemiology and Genetics, National Cancer Institute, NIH, Bethesda, MD 20892, USA.

[75]Department of Medical Epidemiology and Biostatistics, Karolinska Institute, Stockholm 30303, Sweden. [76]Epidemiology Research Program, American Cancer Society, 250 Williams Street, Atlanta, GA 30303, USA. [77]SWOG Statistical Center, Fred Hutchinson Cancer Research Center, Seattle, WA 98109, USA. [78]Australian Prostate Cancer Research Centre-Qld, Institute of Health and Biomedical Innovation and School of Biomedical Science, Queensland University of Technology, Brisbane 4059 Queensland, Australia. [79]Translational Research Institute, Brisbane 4102 Queensland, Australia. [80]Department of Medical Biochemistry and Genetics, Institute of Biomedicine, University of Turku, Turku 20520, Finland. [81]Tyks Microbiology and Genetics, Department of Medical Genetics, Turku University Hospital, Turku 20520, Finland. [82]Division of Nutritional Epidemiology, Institute of Environmental Medicine, Karolinska Institutet, Stockholm 171 77, Sweden. [83]Department of Surgical Sciences, Uppsala University, Uppsala 751 85, Sweden. [84]Division of Cancer Sciences, University of Manchester, Manchester Academic Health Science Centre, Radiotherapy Related Research, Manchester NIHR Biomedical Research Centre, The Christie Hospital NHS Foundation Trust, Manchester M13 9NT, UK. [85]Department of Epidemiology, Harvard T.H Chan School of Public Health, Boston, MA 02115, USA. [86]CeRePP, Tenon Hospital, Paris 75020, France. [87]Sorbonne Université, GRC n°5 ONCOTYPE-URO, Tenon Hospital, Paris 75970, France. [88]Department of Molecular Medicine, Aarhus University Hospital, Aarhus 8000, Denmark. [89]Department of Clinical Medicine, Aarhus University, Aarhus 8000, Denmark. [90]Department of Medical Genetics, Oslo University Hospital, Oslo 0424, Norway. [91]Department of Oncology, Addenbrooke's Hospital, University of Cambridge, Cambridge CB2 0QQ, UK. [92]Cancer Research UK Cambridge Research Institute, Li Ka Shing Centre, Cambridge CB2 0RE, UK. [93]Nuffield Department of Surgical Sciences, University of Oxford, Oxford OX3 7LF, UK. [94]Faculty of Medical Science, John Radcliffe Hospital, University of Oxford, Oxford OX3 9DU, UK. [95]School of Social and Community Medicine, University of Bristol, Bristol BS8 2PS, UK. [96]Cancer Epidemiology Unit, Nuffield Department of Population Health University of Oxford, Oxford OX3 7LF, UK. [97]Department of Surgical Oncology, Princess Margaret Cancer Centre, Toronto M5G 2M9, Canada. [98]Department of Radiation Oncology, Icahn School of Medicine at Mount Sinai, New York, NY 10029, USA. [99]Department of Genetics and Genomic Sciences, Icahn School of Medicine at Mount Sinai, New York, NY 10029, USA. [100]Centre for Molecular Oncology, Barts Cancer Institute, John Vane Science Centre, Queen Mary University of London, London EC1M 6BQ, UK. [101]Cancer Epidemiology & Intelligence Division, The Cancer Council Victoria, Melbourne 3004 Victoria, Australia. [102]Centre for Epidemiology and Biostatistics, Melbourne School of Population and Global Health, The University of Melbourne, Melbourne 3053, Australia. [103]Division of Urologic Surgery, Brigham and Womens Hospital, Boston, MA 02115, USA. [104]Fundación Pública Galega de Medicina Xenómica-SERGAS, Grupo de Medicina Xenómica, CIBERER, IDIS, Santiago de Compostela 15782, Spain. [105]Centre for Research in Environmental Epidemiology (CREAL), Barcelona Institute for Global Health (ISGlobal), Barcelona 60803, Spain. [106]CIBER Epidemiología y Salud Pública (CIBERESP), Madrid 28029, Spain. [107]IMIM (Hospital del Mar Research Institute), Barcelona 08003, Spain. [108]Universitat Pompeu Fabra, Barcelona 08002, Spain. [109]Channing Division of Network Medicine, Department of Medicine, Brigham and Women's Hospital/Harvard Medical School, Boston, MA 02115, USA. [110]Department of Cancer Epidemiology, Moffitt Cancer Center, Tampa 33612, USA. [111]Division of Public Health Sciences, Fred Hutchinson Cancer Research Center, Seattle 98109 Washington, USA. [112]Department of Epidemiology, School of Public Health, University of Washington, Seattle 98195 Washington, USA. [113]International Hereditary Cancer Center, Department of Genetics and Pathology, Pomeranian Medical University, Szczecin 70-001, Poland. [114]Faculty of Health and Medical Sciences, University of Copenhagen, Copenhagen 1165, Denmark. [115]Department of Clinical Biochemistry, Herlev and Gentofte Hospital, Copenhagen University Hospital, Herlev 2900, Denmark. [116]Institute for Human Genetics, University Hospital Ulm, Ulm 89081, Germany. [117]Department of Genitourinary Medical Oncology, The University of Texas MD Anderson Cancer Center, Houston, TX 77030, USA. [118]Cancer Prevention Institute of California, Fremont, CA 94538, USA. [119]Department of Health Research & Policy (Epidemiology) and Stanford Cancer Institute, Stanford University School of Medicine, Stanford, CA 94305, USA. [120]Department of Genetics, Portuguese Oncology Institute of Porto, Porto 4200-072, Portugal. [121]Biomedical Sciences Institute (ICBAS), University of Porto, Porto 4200-072, Portugal. [122]Department of Population Sciences, Beckman Research Institute of the City of Hope, Duarte, CA 91016, USA. [123]Faculty of Medicine and Health Sciences, Basic Medical Sciences, Ghent University, Gent 9000, Belgium. [124]Faculty of Medicine, Department of Surgery, University of Malaya, Kuala Lumpur 50603, Malaysia. [125]Department of Urology, University of Washington, Seattle, WA 98105, USA. [126]Division of Medical Oncology, Urogenital Unit, Department of Oncology, University Hospital Centre Zagreb, 10 000 Zagreb, Croatia. [127]Molecular Medicine Center, Department of Medical Chemistry and Biochemistry, Medical University, Sofia 1431, Bulgaria. [128]Department of Oncology, Cross Cancer Institute, University of Alberta, Edmonton T6G 2R3 Alberta, Canada. [129]Division of Radiation Oncology, Cross Cancer Institute, Edmonton T6G 1Z2 Alberta, Canada. [130]Molecular Endocrinology Laboratory, Department of Cellular and Molecular Medicine, KU Leuven 3000 Leuven, Belgium. [131]Institute of Cancer Sciences, Manchester Cancer Research Centre, University of Manchester, Manchester Academic Health Science Centre, St Mary's Hospital, Manchester M13 9WL, UK. [132]University of California San Diego, Moores Cancer Center, La Jolla, CA 92093, USA. [133]Department of Urology, Erasmus University Medical Center, Rotterdam 3015, the Netherlands. [134]Cancer & Environment Group, Center for Research in Epidemiology and Population Health (CESP), INSERM, University Paris-Sud, University Paris-Saclay, Villejuif 94805, France. [135]Clinical Gerontology Unit, University of Cambridge, Cambridge CB2 2QQ, UK. [136]Division of Genetic Epidemiology, Department of Medicine, University of Utah School of Medicine, Salt Lake City, UT 84108-1266, USA. [137]George E. Wahlen Department of Veterans Affairs Medical Center, Salt Lake City, UT 84148, USA. [138]The University of Surrey, Guildford GU2 7XH Surrey, UK. [139]Department of Laboratory Medicine and Pathology, Mayo Clinic, Rochester, MN 55905, USA

