## [Peer Review File · Nature Communications]

Reviewers' comments:

Reviewer #1 (Remarks to the Author):

Law and colleagues present results of the largest European-specific GWAS meta-analysis of colorectal cancer (CRC). They identify novel loci for CRC at genome-wide significance, and distinct signals of association at known and novel loci, bringing the total number of distinct associations to 79. The lead SNPs at these signals were demonstrated to be enriched in regulatory elements with strong colonic tissue specificity. A range of approaches and data resources were then used to link lead SNPs to potential target genes to provide insight into downstream disease biology.

Overall, the GWAS is well conducted, following standard protocols. The manuscript is nicely written, well structured, and the display items are relevant and of good quality.

Major comment.

The enrichment analyses, and much of the downstream interrogation focuses only on the lead SNPs for each CRC association signal, or a set of SNPs defined to be in "high-LD" with the lead SNP. This approach seems rather naïve to me. It would make much more sense to take account of the relative associations of SNPs at each signal, rather than taking only the lead SNP (which may not be causal) or those at some arbitrary LD threshold with the lead SNP (many of which may have relatively weak associations compared with the lead SNP). FGWAS can be used, for example, to search for enrichment of association signals in specific genomic annotations, either genome-wide or within a limited set of loci, and accounts for the relative strength of associations in the analysis. Formal co-localisation analyses, that take account of the association signals of both the GWAS SNP and eQTL, would also provide a more robust and powerful evidence of a direct link between association signal and target gene.

Minor comments

Line 160. Would be good to specify imputation reference panels used here (even though there are several used).

Line 164. What was λ_{GC} , rather than λ_{1000} ? Since you have used LDSCORE regression, it might make more sense to present the intercept from this analysis to assess residual inflation due to population structure.

Line 170. What was the rationale to report BFDPs? These make prior assumptions that might not be reasonable? How do we decide what we believe based on BFDP? I would suggest removing this.

Line 182. Is it clear WHY the association signals previously reported in Europeans fail to attain even near genome-wide significance in the current study? Is there evidence of heterogeneity in effects at these SNPs? Are they relatively rare in Europeans? Do these previous reports include GWAS that are also part of the current meta-analysis, or are the samples completely non-overlapping?

Line 186. It would be useful to know if the association signals at the nine Asian loci co-localise with those in Europeans – knowing D-prime is high is not sufficient. I would suggest running (approximate) conditional analyses, conditioning the European association signal on the lead SNP reported in Asians. For the Asian loci that do not show any evidence of association in Europeans, are the lead Asian SNPs rare or monomorphic in Europeans?

Line 202. Referring to “known CRC risk SNPs” gives the impression that they are causal, which they will not necessarily be. Better to refer to these as CRC association signals?

Line 247. What is meant by “strong support”? And later, what is meant by “strong/good evidence”?

Line 372. Why restrict to 1:4 matching in UK Biobank?

Line 384. Error with reference.

Line 461. GCTA requires a reference individual-level genotype dataset to approximate LD. What was used for the approximate conditional analyses?

Line 543. LDSCORE regression and LDAK have been used at different times for analyses relating to heritability. They make use of different underlying models of the impact of LD structure. What was the rationale for using different methods for different analyses?

Reviewer #2 (Remarks to the Author):

This manuscript identified 31 new CRC risk loci using data from a colorectal cancer (CRC) GWAS of 34,627 cases and 71,379 controls of European ancestry. This included 12,101 cases and 20,391 controls from 10 published GWAS. In addition, this study found limited support for 8 previously reported loci. The manuscript also provides insights into possible biological mechanism of identified loci using Chi-C, gene expression and CHIP-seq data. Finally, the authors provide estimates of heritability and polygenic risk stratification.

Overall, this is a well written and comprehensive GWAS paper that contributes substantially to our knowledge of the genetic architecture of CRC. The methodology used is appropriate

- It is not clear whether analyses include all previously published GWAS data for CRC since the latest CRC GWAS in subjects of European descent included a total of 36,948 case subjects and 30,864 control (plus additional subjects for replication of top hits) (Schmit et al JNCI 2018). Not sure how this related to the 12,101 cases and 20,391 controls with previous GWAS data used in this analysis. This should be made clearer.
- Without going in detail through the tables and references, it is hard to compare the numbers of previously identified and novel loci. For instance, Schmit et al claimed to have identified 11 novel variants at GW significance level that added to 42 loci previously identified (i.e. 53 loci). However, the current paper talks about 40 previously reported loci at GW significance in Europeans. These comparisons are quite tedious, and it would be helpful if the text could explain more clearly how the numbers quoted here related to previously reported claims.
- The PRS based on 79 GW significant loci is likely to over-estimate risk because of winners' course in effect estimates – this should be addressed in the paper. Although all the data has been used for the discovery of loci, authors could use methods for PRS development that address this problem or identify an external dataset to evaluate the performance of the PRS.
- There are some error messages for references in the text (pages 11-2)

Reviewer #1 (Remarks to the Author):

Law and colleagues present results of the largest European-specific GWAS meta-analysis of colorectal cancer (CRC). They identify novel loci for CRC at genome-wide significance, and distinct signals of association at known and novel loci, bringing the total number of distinct associations to 79. The lead SNPs at these signals were demonstrated to be enriched in regulatory elements with strong colonic tissue specificity. A range of approaches and data resources were then used to link lead SNPs to potential target genes to provide insight into downstream disease biology.

Overall, the GWAS is well conducted, following standard protocols. The manuscript is nicely written, well structured, and the display items are relevant and of good quality.

1.1 The enrichment analyses, and much of the downstream interrogation focuses only on the lead SNPs for each CRC association signal, or a set of SNPs defined to be in “high-LD” with the lead SNP. This approach seems rather naïve to me. It would make much more sense to take account of the relative associations of SNPs at each signal, rather than taking only the lead SNP (which may not be causal) or those at some arbitrary LD threshold with the lead SNP (many of which may have relatively weak associations compared with the lead SNP). FGWAS can be used, for example, to search for enrichment of association signals in specific genomic annotations, either genome-wide or within a limited set of loci, and accounts for the relative strength of associations in the analysis. Formal co-localisation analyses, that take account of the association signals of both the GWAS SNP and eQTL, would also provide a more robust and powerful evidence of a direct link between association signal and target gene.

Response: We used SMR (Summary-data-based Mendelian Randomization, Zhu et al. 2016 Nature Genetics) to link genetic variation at a region to expression of a target gene. The methodology was formulated by a well-established statistical genetics group (led by Peter Visscher and Jian Yang) and is a widely adopted procedure which takes into account both the GWAS association statistics and the significance of the eQTL across the region. To supplement this analysis we have implemented CHi-C to provide direct evidence of a relationship between genomic region and target gene. In view of these analyses we feel that implementing FGWAS is not necessary as most aspects of FGWAS have been covered (e.g. the overall enrichment for certain genomic features). Furthermore, FGWAS only considers one functional variant per region.

1.2 Line 160. Would be good to specify imputation reference panels used here (even though there are several used).

Response: The details of the imputation panels used are described in the Methods section. We have clarified the text to indicate the primary reference panels used.

1.3 Line 164. What was λ_{GC} , rather than λ_{1000} ? Since you have used LDSCORE regression, it might make more sense to present the intercept from this analysis to assess residual inflation due to population structure.

Response: We feel it is entirely appropriate to report λ_{1000} for the meta-analysis as λ_{GC} is heavily influenced by sample size. However, we have stated in the Methods the λ_{GC} for each of the contributing GWAS, and none of these showed significant genomic inflation. We also include reference to the individual λ_{GC} values in the Results. In light of the reviewer’s later comment about LDSCORE (1.12), we have removed the statement to avoid confusion.

1.4 Line 170. What was the rationale to report BFDPs? These make prior assumptions that might not be reasonable? How do we decide what we believe based on BFDP? I would suggest removing this.

Response: The aim of using the BFDP is to reduce the number of potential false positive results. It has been shown that naïve use of P -values as a measure of association for SNPs can result in the over-estimation of their significance (Bigdeli et al, Bioinformatics. 2016; Wakefield, Am J Hum

Genet. 2007). Due to the large number of tests, many SNPs will attain very low P -values even under the null hypothesis of no association between trait and genotypes. Additionally, a common misconception is to view P -values as the probability of the null hypothesis given the observed test statistic, when they are the probability of the statistic given the hypothesis. By using the BFDP we are able to assess the probability of the hypothesis given the data, and derive likely priors from the data. We believe that inclusion of BFDP values is in line with much current thinking and hence wish to retain this.

1.5 Line 182. Is it clear WHY the association signals previously reported in Europeans fail to attain even near genome-wide significance in the current study? Is there evidence of heterogeneity in effects at these SNPs? Are they relatively rare in Europeans? Do these previous reports include GWAS that are also part of the current meta-analysis, or are the samples completely non-overlapping?

Response: With the exception of a minority of samples (CCFR, CORSA, some of UKBB), the studies in our analysis and that of Peters and co-workers are independent. In general, the SNPs that failed to replicate were identified in studies that were not part of this GWAS. Without direct access to these data, we can only speculate as to possible reasons for disparities. While the previous studies were reported to be in individuals of European descent, one explanation is some effect due to undetected population heterogeneity. Alternatively, the disparity in study findings may have arisen because of differences in the criteria for retaining imputed SNPs in analyses. While we have imposed a stringent INFO score of ≥ 0.8 for retaining imputed SNP genotypes, we note that other studies have included SNPs with much lower INFO scores (typically ~ 0.3).

1.6. Line 186. It would be useful to know if the association signals at the nine Asian loci co-localise with those in Europeans – knowing D -prime is high is not sufficient. I would suggest running (approximate) conditional analyses, conditioning the European association signal on the lead SNP reported in Asians. For the Asian loci that do not show any evidence of association in Europeans, are the lead Asian SNPs rare or monomorphic in Europeans?

Response: After performing a conditional analysis on the reported Asian SNPs in the European signals, five of the nine European SNPs were identified as independent signals ($P_{\text{conditional}} < 5 \times 10^{-8}$). These findings are in line with the respective R^2 values.

For the Asian loci that did not show any evidence of association in Europeans, there is no obvious unifying explanation. In some cases, the risk allele frequency (RAF) is very different between populations (e.g. rs11064437 is monomorphic in Europeans), although there are also cases where SNPs with different RAFs did replicate (e.g. rs704017 where the RAF in Europeans is almost double that in Asians). These results are now included in Supplementary Table 4.

1.7. Line 202. Referring to “known CRC risk SNPs” gives the impression that they are causal, which they will not necessarily be. Better to refer to these as CRC association signals?

Response: We acknowledge this point and have revised our text accordingly.

1.8. Line 247. What is meant by “strong support”? And later, what is meant by “strong/good evidence”?

Response: We acknowledge that this may be ambiguous and have revised our text accordingly, qualifying statements about the strength of support with P -values.

1.9. Line 372. Why restrict to 1:4 matching in UK Biobank?

Response: Our rationale for 1:4 case-to-controls was that a greater number of controls does not yield a significant increase in statistical power for the range of risk alleles frequencies we had pre-specified in our analysis (i.e. $>0.5\%$); Indeed a 1:4 ratio has been reported to be optimal for 80%

statistical power previously (Hong and Park, Genomics Inform. 2012). Furthermore, we were advised to use a 1:4 case:control ratio by the UKBB Research Access team when we applied for the UK Biobank data. We do, however acknowledge that for the identification of rare and low frequency variants, larger numbers of controls may be desirable, although this brings into question how well this class of allele can be imputed.

1.10. Line 384. Error with reference.

Response: Typographical error corrected.

1.11. Line 461. GCTA requires a reference individual-level genotype dataset to approximate LD. What was used for the approximate conditional analyses?

Response: We used the NSCCG-Oncoarray data as the reference data. The validity of using one of the large participating cohorts as the reference is suggested by the authors of GCTA (<http://gcta.freeforums.net/thread/178/conditional-joint-analysis-using-summary>)

1.12. Line 543. LDSCORE regression and LDAK have been used at different times for analyses relating to heritability. They make use of different underlying models of the impact of LD structure. What was the rationale for using different methods for different analyses?

Response: As per 1.3, we now omit the LDSCORE analysis here.

Reviewer #2 (Remarks to the Author):

This manuscript identified 31 new CRC risk loci using data from a colorectal cancer (CRC) GWAS of 34,627 cases and 71,379 controls of European ancestry. This included 12,101 cases and 20,391 controls from 10 published GWAS. In addition, this study found limited support for 8 previously reported loci. The manuscript also provides insights into possible biological mechanism of identified loci using Chi-C, gene expression and ChIP-seq data. Finally, the authors provide estimates of heritability and polygenic risk stratification.

Overall, this is a well written and comprehensive GWAS paper that contributes substantially to our knowledge of the genetic architecture of CRC. The methodology used is appropriate

2.1. It is not clear whether analyses include all previously published GWAS data for CRC since the latest CRC GWAS in subjects of European descent included a total of 36,948 case subjects and 30,864 control (plus additional subjects for replication of top hits) (Schmit et al JNCI 2018). Not sure how this related to the 12,101 cases and 20,391 controls with previous GWAS data used in this analysis. This should be made clearer.

Response: With the exception of samples from CCFR, CORSA, and some of UKBB (primarily only used in the replication phase of Schmit et al JNCI 2018) the studies are independent. The overlap accounts for ~20% of cases in our analysis.

2.2. Without going in detail through the tables and references, it is hard to compare the numbers of previously identified and novel loci. For instance, Schmit et al claimed to have identified 11 novel variants at GW significance level that added to 42 loci previously identified (i.e. 53 loci). However, the current paper talks about 40 previously reported loci at GW significance in Europeans. These comparisons are quite tedious, and it would be helpful if the text could explain more clearly how the numbers quoted here related to previously reported claims.

Response: The main discrepancy is a result of Schmit et al combining all CRC susceptibility together, regardless of the original population, whereas we separated the loci based on European

or Asian sample populations. We have clarified the text. In addition, some of the loci reported by Schmit et al were not genome-wide significant ($P < 5 \times 10^{-8}$) in the original findings, e.g. rs719725 at 9p24 (Zanke et al, Nat Gen, 2007) was originally reported as $P = 1.32 \times 10^{-5}$.

2.3. The PRS based on 79 GW significant loci is likely to over-estimate risk because of winners' course in effect estimates – this should be addressed in the paper. Although all the data has been used for the discovery of loci, authors could use methods for PRS development that address this problem or identify an external dataset to evaluate the performance of the PRS.

Response: We acknowledge the issues of the winner's curse in estimating effect size and have now performed a correction using the FDR Inverse Quantile Transformation (FIQT) method (Bigdeli et al. Bioinformatics 2016). We do not have access to additional large datasets to validate the findings, but also believe such an additional analysis is beyond the remit of the current paper.

2.4. There are some error messages for references in the text (pages 11-2)

Response: Errors in these references have been corrected.

REVIEWERS' COMMENTS:

Reviewer #1 (Remarks to the Author):

Overall, the authors have adequately addressed my comments. I appreciate the use of SMR as a well established method for linking genetic variation to local gene expression. Whilst the authors have used a published method for assessing enrichment of CRC SNPs with TF binding across loci, I believe that focussing on lead SNPs and those in strong LD with the lead SNPs is sub-optimal, and hence my suggestion to use fGWAS. I agree that in a basic fGWAS analysis that a single causal SNP is present at any given locus, it is also possible to incorporate results from approximate conditional analyses. It is not clear to me that using lead SNPs and those in strong LD with the lead SNP addresses this issue.

Reviewer #2 (Remarks to the Author):

The authors have adequately addressed my comments and suggestions for revision

REVIEWERS' COMMENTS:

Reviewer #1 (Remarks to the Author):

Overall, the authors have adequately addressed my comments. I appreciate the use of SMR as a well established method for linking genetic variation to local gene expression. Whilst the authors have used a published method for assessing enrichment of CRC SNPs with TF binding across loci, I believe that focussing on lead SNPs and those in strong LD with the lead SNPs is sub-optimal, and hence my suggestion to use fGWAS. I agree that in a basic fGWAS analysis that a single causal SNP is present at any given locus, it is also possible to incorporate results from approximate conditional analyses. It is not clear to me that using lead SNPs and those in strong LD with the lead SNP addresses this issue.

We do not believe that using fGWAS to perform a TF binding enrichment would substantially change the results from those we have derived using another published methodology. While the strong LD restriction may be conservative, we assert that such an analysis is appropriate. In addition, it is consistent with all the other analyses that were performed in the manuscript. Furthermore, fGWAS divides the genome into arbitrarily sized blocks, which ignores LD structure. All analyses were performed including any independent signals as determined by a conditional analysis, so these data have indeed been taken into account.

Reviewer #2 (Remarks to the Author):

The authors have adequately addressed my comments and suggestions for revision

We thank the reviewer for their consideration.